# Rab7a is an enhancer of TPC2 activity regulating melanoma progression through modulation of the GSK3β/β-Catenin/MITF-axis

Carla Abrahamian [1,2,11], Rachel Tang[1,11], Rebecca Deutsch[1,11], Lina Ouologuem[3,11], Eva-Maria Weiden [1], Veronika Kudrina [1], Julia Blenninger[3], Julia Rilling[3], Colin Feldmann[4], Solveig Kuss [5], Youli Stepanov[6], Anna Scotto Rosato[1], Guadalupe T. Calvo[7], Maria S. Soengas [7], Doris Mayr[5], Thomas Fröhlich [6], Thomas Gudermann[1], Martin Biel[3], Christian Wahl-Schott[4], Cheng-Chang Chen [8,9], Karin Bartel [3] ✉ & Christian Grimm [1,10] ✉

Melanoma arising from pigment-producing melanocytes is the deadliest form of skin cancer. Extensive ultraviolet light exposure is a major cause of melanoma and individuals with low levels of melanin are at particular risk. Humans carrying gain-of-function polymorphisms in the melanosomal/endolysosomal two-pore cation channel TPC2 present with hypopigmentation, blond hair, and albinism. Loss of TPC2 is associated with decreased cancer/melanoma proliferation, migration, invasion, tumor growth and metastasis formation, and TPC2 depleted melanoma cells show increased levels of melanin. How TPC2 activity is controlled in melanoma and the downstream molecular effects of TPC2 activation on melanoma development remain largely elusive. Here we show that the small GTPase Rab7a strongly enhances the activity of TPC2 and that effects of TPC2 on melanoma hallmarks, in vitro and in vivo strongly depend on the presence of Rab7a, which controls TPC2 activity to modulate GSK3β, β-Catenin, and MITF, a major regulator of melanoma development and progression.

Rab7 proteins are small GTPases, which can bind and hydrolyze guanosine triphosphate (GTP). They belong to a large protein family of more than 70 members in mammalian genomes and play fundamental roles in e.g., intracellular trafficking, vesicle formation, vesicle movement, and membrane fusion. Rabs are localized to the cytoplasmic face of intracellular organelles and vesicles and are highly selective in their subcellular localization. Two Rab7 proteins are known, Rab7a and Rab7b, which share about 50% sequence similarity.

[1]Walther Straub Institute of Pharmacology and Toxicology, Faculty of Medicine, Ludwig-Maximilians-University, Munich, Germany. [2]Department of Cardiology, German Heart Centre Munich, Technical University of Munich, Munich, Germany. [3]Department of Pharmacy, Ludwig-Maximilians-University, Munich, Germany. [4]Institute of Cardiovascular Physiology and Pathophysiology, Faculty of Medicine, Ludwig-Maximilians-University, Munich, Germany. [5]Institute of Pathology, Faculty of Medicine, Ludwig-Maximilians-University, Munich, Germany. [6]Laboratory for Functional Genome Analysis LAFUGA, Gene Center, Ludwig-Maximilians-University, Munich, Germany. [7]Melanoma Laboratory, Molecular Pathology Programme, Centro Nacional de Investigaciones Oncológicas (Spanish National Cancer Research Centre), Madrid, Spain. [8]Department of Clinical Laboratory Sciences and Medical Biotechnology, College of Medicine, National Taiwan University, Taipei, Taiwan. [9]Department of Laboratory Medicine, National Taiwan University Hospital, Taipei, Taiwan. [10]Immunology, Infection and Pandemic Research IIP, Fraunhofer Institute for Translational Medicine and Pharmacology ITMP, Munich/Frankfurt, Germany. [11]These authors contributed equally: Carla Abrahamian, Rachel Tang, Rebecca Deutsch, Lina Ouologuem. ✉e-mail: karin.bartel@cup.uni-muenchen.de; christian.grimm@med.uni-muenchen.de

Rab7a localizes to late endosomes/lysosomes (LE/LY) and controls vesicular transport from early endosomes (EE) to LE/LY in the endocytic pathway. Rab7a plays a fundamental role not only for trafficking and degradation of many signaling receptors e.g., EGF/EGFR and adhesion molecules, but also for biogenesis, positioning, and motility of lysosomes as well as auto- and phagolysosomes[1–4]. Rab7a further plays key roles in cell survival, growth, differentiation, migration, autophagy and apoptosis. Modulation of Rab7a activity affects a number of disease pathologies including neuropathies and neurodegenerative diseases such as Charcot-Marie-Tooth type 2B, hereditary sensory neuropathy type 1, and Niemann Pick type C1 (NPC1), infectious diseases, and cancer, including melanoma[5–10]. Thus, Rab7a is e.g., associated with poor prognosis of gastric cancer and promotes proliferation, invasion, and migration of gastric cancer cells. Knockdown of Rab7a suppresses the proliferation, migration, and xenograft tumor growth of breast cancer cells and high Rab7a expression is an indicator of a higher risk of metastasis in early melanoma patients. Furthermore, in melanoma cells Rab7a levels are significantly elevated compared to normal skin melanocytes, impacting melanoma proliferation and invasion[5,9,11,12]. Similar to Rab7a, the $Na^+$ and $Ca^{2+}$ permeable cation channel TPC2 in LE/LY and melanosomes of melanocytes has been described as an important regulator of endolysosomal trafficking, with EGF/EGFR, LDL cholesterol, or PDGF accumulating in TPC2 knockout cells[13–16]. In analogy to TPC2 inhibition, knockdown or knockout, Rab7a knockdown in NPC1 cells exacerbates cholesterol accumulation[5,17,18]. Inhibition, knockdown or knockout of TPC2 also results in reduced proliferation, migration, and invasion as well as reduced tumor growth, metastasis formation and VEGF-induced angiogenesis in different types of cancer, including melanoma[16,19–23]. Besides, TPC2 affects also melanin production and pigmentation in melanocytes and melanoma cells[19,24,25] (due to its expression and activity in melanosomes, which are lysosome related organelles). Several human gain-of-function (GOF) polymorphisms, $TPC2^{M484L}$, $TPC2^{G734E}$, and recently $TPC2^{R210C}$ were found to result in hypopigmentation, blond hair color and dominant albinism[26–28].

In proteomics studies Rab7a was identified as a potential interaction partner of TPC2 and interaction between TPC2 and Rab7a involving the N-terminus of TPC2 (residues 33-37) was postulated[29]. However, a detailed analysis of how Rab7a affects TPC2 channel activity and function, especially by direct electrophysiological means is missing and the physiological or pathophysiological consequences associated with this remain largely elusive. By using endolysosomal patch-clamp electrophysiology and GCaMP based $Ca^{2+}$ imaging experiments we show here that Rab7a strongly increases TPC2 activity. TPC2 is physically interacting with Rab7a, confirmed by co-immunoprecipitation experiments and by FRET (Fluorescence Resonance Energy Transfer) experiments for real-time and quantitative information about Rab7a and TPC2 protein interaction in living cells with high spatial and temporal resolution. The modulatory effect of Rab7a on TPC2 activity is further examined by acute application of a small molecule inhibitor of Rab7, which instantly reverses the effect on TPC2 activity. Expression of Rab7a in different cancer types strongly correlates with the expression of TPC2, with expression of both proteins being particularly high in melanoma cells. Rab7a, the major Rab7 protein in melanoma cells (Rab7b is expressed at much lower levels compared to Rab7a) is postulated here to control melanoma proliferation, growth, and invasiveness through TPC2 activity regulation, specifically in melanoma cells which express high levels of MITF (microphthalmia-associated transcription factor), a known master regulator of melanocytes and melanoma development and progression. Mechanistically, TPC2 activation enhanced by Rab7a increases the endolysosomal degradation of GSK3β, contained in distinct destruction complexes[30], thus preventing proteasomal degradation of MITF. This finding elucidates the mechanism by which an endolysosomal cation channel (TPC2) in concert with a small GTPase (Rab7a)

controls endolysosomal degradation of GSK3β, a key regulator of β-Catenin and MITF expression.

## Results

### Expression of Rab7a correlates with TPC2 in human melanoma cell lines

We screened multiple human cancer cell lines including hepatocellular carcinoma (HCC; Huh7, HepG2), breast (MDA-MB-231, MCF-7, SK-BR-3), ovarian (SKOV3), cervical (HeLa), colon (Caco-2), and pancreatic (Panc-1) cancer lines as well as glioma (U87MG) and melanoma (SK-MEL-5, SK-MEL-2, SK-MEL-28, A375, SK-MEL-29, SK-MEL-19, SK-MEL-147, SK-MEL-103, UACC-62, UACC-257, MNT-1, CHL-1) lines for Rab7a mRNA expression and found particular high expression levels in the majority of melanoma lines (Fig. 1a). Western blot (WB) results assessing Rab7a expression in different melanoma lines and the breast cancer line MDA-MB-231 as control were found to correlate well with the expression levels in the qPCR dataset (Fig. 1b) i.e., lines with high transcript levels in qPCR also showed high expression levels in WB and vice versa. Compared to Rab7a expression of Rab7b was minimal in all tested cancer lines (Fig. 1c). Expression of Rab7a was found to correlate with the expression of the human two-pore channel TPC2 (predominantly found in LE/LY like Rab7a), but not with the TPC2-related channel TPC1 (predominantly found in early endosomes (EE)) (Fig. 1d). Furthermore, mRNA expression of TPC2 was found to be much higher compared to TPC1 in most melanoma lines while in other cancer lines expression levels were generally lower compared to the melanoma lines (Fig. 1e). We confirmed these findings by measuring TPC2 currents using endolysosomal patch-clamp electrophysiology in different cancer lines, revealing that currents are much larger in SK-MEL-5 (melanoma line with high TPC2/Rab7a levels in qPCR experiments) as compared to HeLa or MDA-MB-231 cells (low TPC2/Rab7a levels in qPCR experiments) (Fig. S1a). Finally, higher levels of Rab7 were detected in the majority of human metastatic melanoma compared to healthy control tissue samples (Fig. 1f, g). In sum, these data demonstrate high expression levels of both TPC2 and Rab7a in melanoma and/or melanoma lines while expression levels in other cancer types or control tissue were generally lower.

### Rab7a physically interacts with TPC2

It has been reported previously that Rab7a interacts with TPC2 via the N-terminus of the latter[29]. We confirmed interaction between Rab7a and TPC2 here by an extended Two-hybrid FRET and co-immunoprecipitation analysis. In FRET experiments, TPC2 coexpression with Rab7a resulted in 27% maximum FRET efficiency (Fig. 1h). FRET efficiency reached 40% when pretreated with apilimod, which results in enlarged endolysosomes as used for patch-clamp. Coexpression with $Rab7a^{Q67L}$, a constitutively active mutant, resulted in 51% FRET efficiency after apilimod treatment while coexpression with the dominant negative Rab7a variant $Rab7a^{T22N}$ yielded only 10% FRET efficiency under these conditions (Fig. S1b). Coexpression with Rab5 (an EE marker showing less colocalization with TPC2) reached only 13% FRET efficiency (Fig. S1c). As further controls we determined the FRET efficiency for TPC1 with Rab5 (= positive control, 29%) and TPC1 with Rab7a (= negative control, 6%) (Figs. 1h, i and S1d). In addition to FRET, we also performed co-immunoprecipitation experiments, which likewise confirmed interaction of TPC2 with Rab7a (Fig. S1e). In sum, these data corroborate a physical interaction of TPC2 but not TPC1 with Rab7a.

### Rab7a functionally interacts with TPC2

To test whether Rab7a functionally interferes with TPC2, endolysosomal patch-clamp and GCaMP based $Ca^{2+}$ imaging experiments were performed (Fig. 2). PI(3,5)P$_2$, TPC2-A1-P (mimicking the effect of PI(3,5)P$_2$[31]), and TPC2-A1-N (mimicking the effect of NAADP[31]) were used as agonists to assess TPC2 activity. Endolysosomal patch-clamp experiments were performed using first HEK293 cells overexpressing either

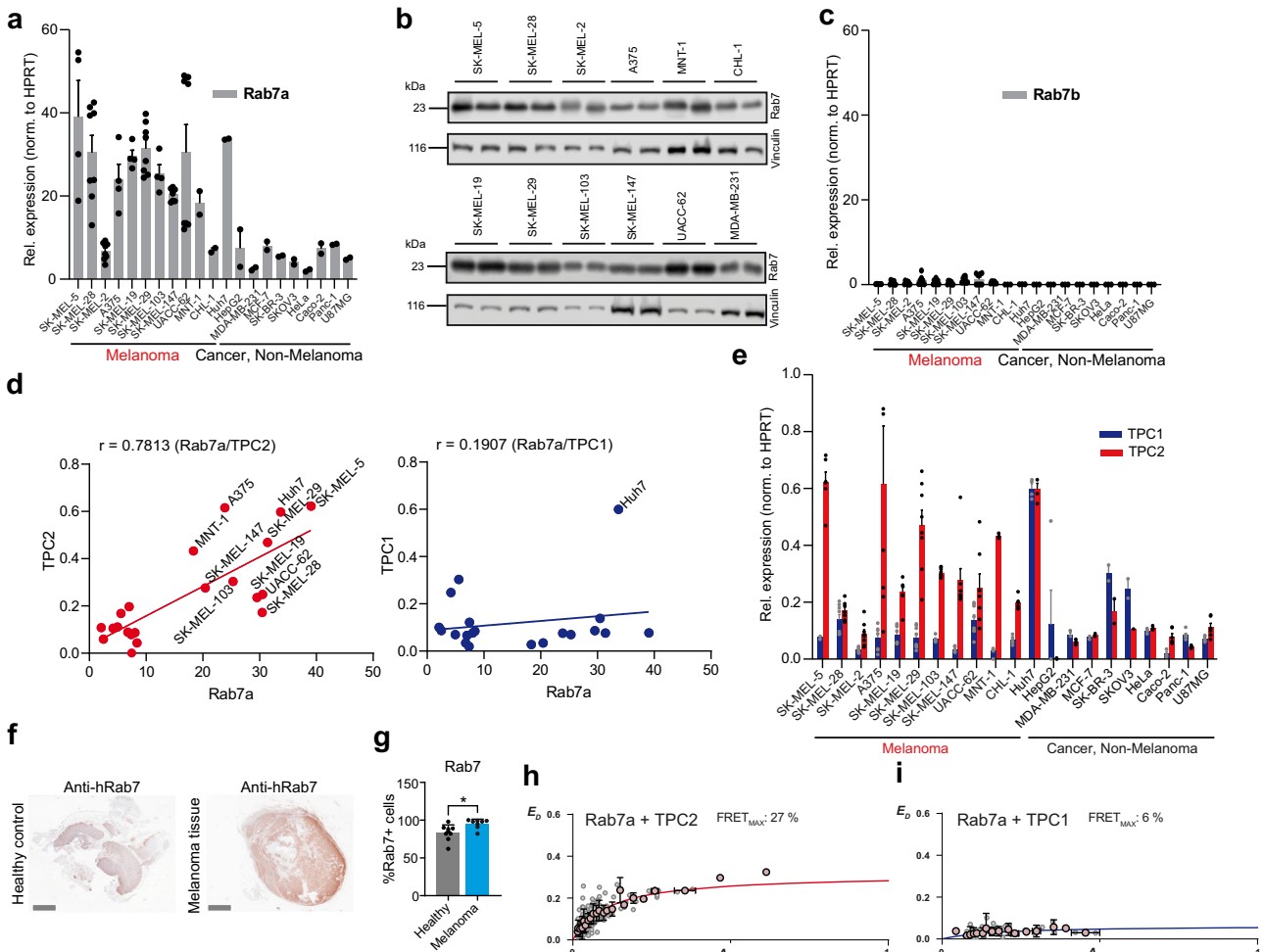

**Fig. 1 | Expression of Rab7a in melanoma versus non-melanoma cancer cells, correlation with TPC2 expression, and interaction of Rab7a with TPC2. a** Gene expression profile (qPCR) of Rab7a in human melanoma lines compared to different non-melanoma cancer lines. Error bars represent mean values ± SEM. Data points represent biological replicates. **b** Protein levels corroborating mRNA expression data in melanoma lines (compared to the breast cancer line MDA-MB-231). Shown are biological replicates, each. **c** Gene expression profile (qPCR) of Rab7b in human melanoma lines compared to different non-melanoma cancer lines. Data points show biological replicates. Error bars represent mean values ± SEM. Data points represent biological replicates. **d** Melanoma lines showing strong expression correlation between Rab7a and TPC2 but not TPC1. **e** Gene expression profile (qPCR) showing relative expression of TPCs in human melanoma lines

compared to different non-melanoma cancer lines. Error bars represent mean values ± SEM. Data points represent biological replicates. **f** Representative images of sections from healthy lymphnode (female, abdomen) and melanoma lymphnode metastasis (male, left forefoot) samples stained with hRab7 antibody (IHC). Scale bars = 5 mm. **g** IHC evaluation was carried out considering the percentage of stained tumor cells. Statistical significance was assessed by two-tailed unpaired $t$-test, *$p = 0.0126$ (mean ± SD, $n = 10$, each). One dot corresponds to one independent human donor. **h**, **i** FRET experiments showing FRET efficiencies in HEK293 cells expressing hTPC2^WT or hTPC1^WT with hRab7a^WT ($n = 163$ and $60$ biological replicates, respectively, error bars are SEM). Source data are provided as Source Data file.

TPC2 alone or coexpressing TPC2 and Rab7a^WT, the constitutively active mutant Rab7a^Q67L, or the dominant negative mutant Rab7a^T22N (Fig. 2a–d). These experiments revealed that Rab7a^WT or Rab7a^Q67L (GTP-logged) coexpression strongly enhanced TPC2 activity, independent of the type of agonist applied, while coexpression with Rab7a^T22N (GDP-logged) or TPC2 expression alone showed significantly smaller current densities when activated with the respective agonists (Fig. 2a–d). These findings were subsequently confirmed in GCaMP-based Ca²⁺ imaging experiments (Fig. 2e). In addition, GTP preincubation for several minutes further enhanced TPC2 channel activity after agonist (TPC2-A1-P) application in endolysosomal patch-clamp experiments, suggesting an effect on TPC2 via endogenously expressed Rab7a and supporting the notion that GTP increases the amount of endogenous GTP-logged Rab7a to modulate TPC2. These data together with the FRET data strongly imply that the GTP-logged active form of Rab7a is necessary for TPC2 activity regulation (Fig. S2). In contrast

to TPC2, coexpression of TRPML1, another endolysosomal Ca²⁺/Na⁺ release channel (also highly expressed in melanoma), with Rab7a^WT or mutants showed no effect (Fig. S3). TRPML1 was stimulated in endolysosomal patch-clamp experiments with either a synthetic, TRPML1 isoform-selective small molecule agonist, ML1-SA1[32] or with the endogenous agonist PI(3,5)P₂ (Fig. S3a–d). The effects were completely blocked by the TRPML1-selective blocker EDME[33].

Next, it was tested whether acute inhibition with the commercially available small molecule Rab7 inhibitor CID1067700 affects TPC2 activity. HEK293 cells coexpressing TPC2 and Rab7a^WT showed a dose-dependent, instant reduction of the Rab7a^WT enhanced channel activity after application of CID1067700. Similar observations were made in cells expressing TPC2 alone (blocking endogenous Rab7) (Fig. 3a–c). In sum, our data suggest that the effect of Rab7a on TPC2 activity is an acute effect and that it is independent of the TPC2 activation mode.

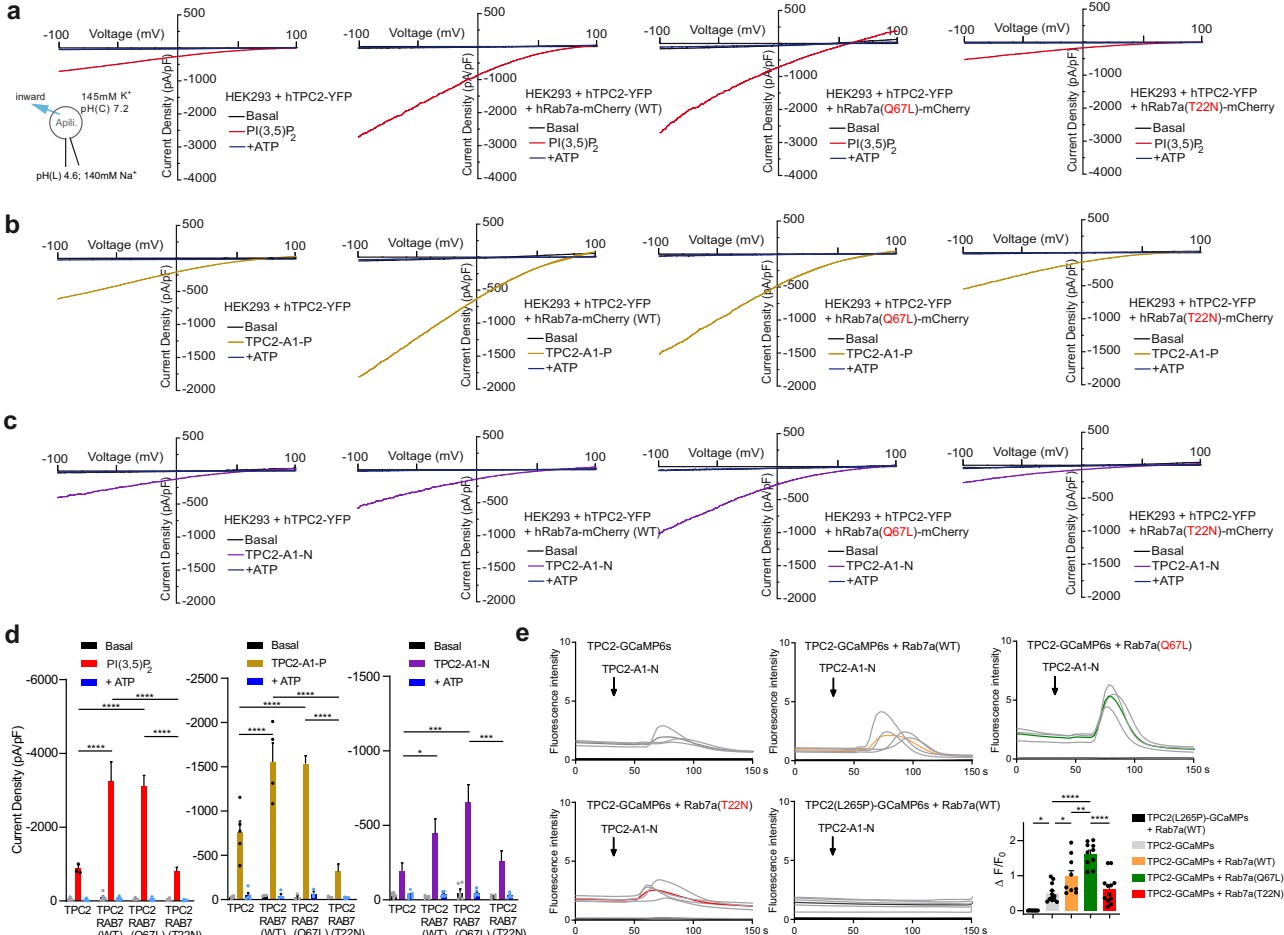

**Fig. 2 | Effect of Rab7a on TPC2 activity. a** Effect of PI(3,5)P$_2$ in endolysosomal vesicles coexpressing human TPC2 and Rab7a or mutant variants of Rab7a. Shown are representative current density-voltage relationships from −100 to +100 mV with basal currents in black, 1 μM PI(3,5)P$_2$ activated currents in red and ATP (1 mM) blocked currents in blue, measured from apilimod-treated, enlarged endolysosomal vesicles expressing either hTPC2$^{WT}$, hTPC2$^{WT}$ + hRab7a$^{WT}$, hTPC2$^{WT}$ + hRAB7a$^{Q67L}$ (constitutively active Rab7a) or hTPC2$^{WT}$ + hRAB7a$^{T22N}$ (dominant negative Rab7a). **b**, **c** Analogous experiments for the lipophilic small molecule agonists of TPC2 TPC2-A1-P and TPC2-A1-N (10 μM, each). **d** Statistical summary of data comprising average current densities (mean ± SEM) at −100 mV measured in endolysosomal patch-clamp experiments as shown in (**a–c**). Each dot on the bar graph represents a single current density value measured from one endolysosome ($n$ = 3–9). Data were tested for statistical significance with one-way ANOVA test followed by Tukey's post-test (*$p < 0.0357$, ***$p < 0.001$,

****$p < 0.0001$). **e** Representative GCaMP6s traces, with mean value curves highlighted in bold (color coded), each. Bar chart: Maximal change in fluorescence after application of TPC2 agonist TPC2-A1-N (mean ± SEM). Change in GCaMP6s fluorescence ($\Delta F$) was normalized to baseline value ($\Delta F/F0$), each. The baseline value ($F0$) was acquired by averaging fluorescence from a 30 s recording before addition of compound. One dot corresponds to one experiment with 3–7 transfected cells, each. TPC2-GCaMP6s ($n$ = 12), TPC2-GCaMP6s + Rab7 WT ($n$ = 9), TPC2-GCaMP6s + Rab7 Q67L ($n$ = 10), TPC2-GCaMP6s + Rab7 T22N ($n$ = 12), L265P-TPC2-GCaMP6s + Rab7 WT ($n$ = 8). Statistical significance was determined via one-way ANOVA followed by Bonferroni multiple comparisons test (p (L265P TPC2 + Rab7a vs. WT TPC2) = 0.0356, p (WT TPC2 vs. TPC2 + Rab7a) = 0.0261, p (WT TPC2 vs. TPC2 + QL Rab7a) < 0.0001, p (TPC2 + Rab7a vs. TPC2 + QL Rab7a) = 0.0037, p (TPC2 + Rab7a vs. TPC2 + TN Rab7a) < 0.0001). Source data are provided as Source Data file.

## Knockout of Rab7a results in a reduction of endogenous TPC2 currents in SK-MEL-5 melanoma cells

Expression levels (mRNA) of Rab7a and TPC2 are particularly high in melanoma cells including SK-MEL-5 cells, which were chosen to generate knockout (KO) lines for both genes using CRISPR/Cas9 strategies (Fig. 3d–m). TPC2 KO was confirmed in three independent lines by genotyping, endolysosomal patch-clamp and qPCR experiments (Figs. S4 and 3d–g; see also Yuan et al.[34]). To demonstrate that Rab7a affects the activity of TPC2 in the endogenous expression system, we next generated a Rab7a SK-MEL-5 KO line. WB and qPCR analyses of the Rab7a KO line (clone C1x17) demonstrated strongly reduced RNA levels and absent protein, confirming the genotyping results (Figs. 3h–j and S4). The electrophysiological analysis revealed absence of TPC2 activity in TPC2 KO and significantly reduced activity in Rab7a KO compared to WT SK-MEL-5 cells (Fig. 3k, l), corroborating the results obtained from overexpressing HEK293 presented in Fig. 2. By contrast,

currents measured from endolysosomes isolated form Rab7a KO SK-MEL-5 cells versus WT SK-MEL-5 cells showed no differences when activated with the synthetic, TRPML1 isoform-selective small molecule agonist ML1-SA1[32] (Fig. S3e–g), confirming the data obtained for TRPML1 in the heterologous expression system (Fig. S3a–d). In sum, our results suggest that Rab7a enhances endogenous TPC2 activity while loss of Rab7a reduces it. In contrast to TPC2, endogenous TRPML1 was not modulated in its activity by Rab7a and the effect on TPC2 was not due to changes in expression as expression levels (qPCR) of TPC2 were unchanged in WT compared to Rab7a KO cells (Fig. 3m).

## Migration, invasion and proliferation in TPC2 and Rab7 knockout and knockdown melanoma cells

As reported previously, knockout or knockdown of TPC2 in different cancer cell lines results in reduction of migration, invasion, and

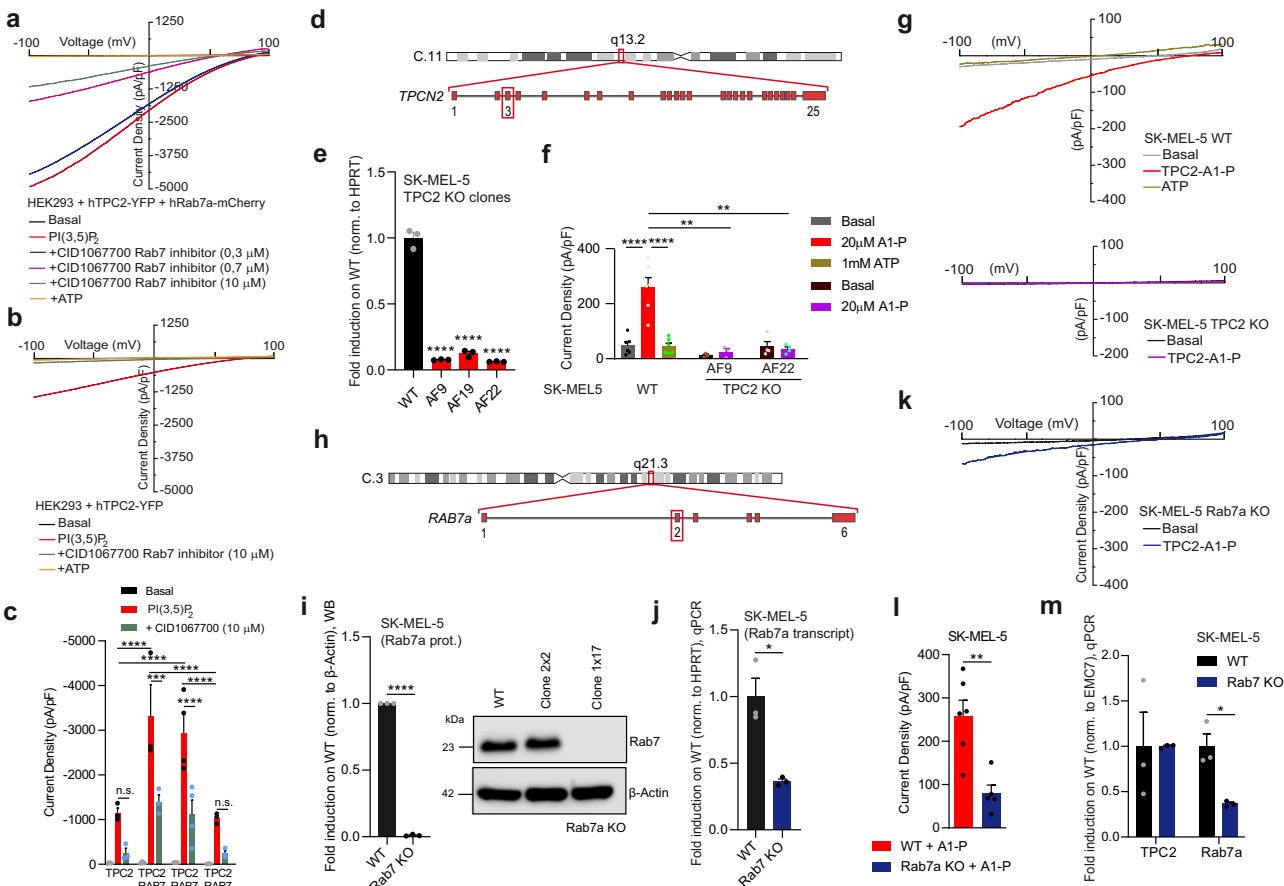

**Fig. 3 | Effect of Rab7 inhibitor on TPC2 activity and physical interaction of Rab7a with TPC2. a, b** Inhibition of PI(3,5)P$_2$ evoked currents in endolysosomes (EL), expressing hTPC2 alone or with Rab7a, using the Rab7-inhibitor CID1067700. Shown are representative current density-voltage relationships of enlarged EL, expressing hTPC2$^{WT}$ + hRab7$^{WT}$ or hTPC2$^{WT}$ alone, activated with 1 μM PI(3,5)P$_2$ followed by application of CID1067700 (diff. conc.) and 1 mM ATP (max. effect). **c** Statistical summary of data as shown in (**a, b**) at −100 mV. Each dot represents a single current density value measured from one EL. Data were tested for statistical significance with one-way ANOVA test followed by Tukey's post-test (***$p < 0.001$, ****$p < 0.0001$, $n = 3$). **d** Cartoon showing CRISPR/Cas9 strategy to knockout *TPCN2* in the SK-MEL-5 cell line. **e** qPCR data showing relative expression of TPC2 in WT and TPC2 KO SK-MEL-5 ($n = 3$). **f** Statistical summary of data (average current densities at −100 mV) as shown in (**g**) ($n = 6$). **g** Representative current density-voltage relationships from −100 to +100 mV showing basal, TPC2-A1-P (20 μM) activated and ATP (1 mM) blocked currents.

**h** Cartoon showing CRISPR/Cas9 strategy to knockout Rab7a in SK-MEL-5. **i** Western blot data showing Rab7a protein levels in WT and Rab7a KO SK-MEL-5 clone C1x17. Clone C2x2 showed no reduction in expression and was not further used ($n = 3$). **j** qPCR data depicting transcript levels of Rab7a KO SK-MEL-5 clone C1x17, compared to WT ($n = 3$). Representative current density-voltage relationships from −100 to +100 mV showing basal and TPC2-A1-P activated currents, measured in SK-MEL-5 Rab7a KO cells from enlarged ELs (**k**) and corresponding statistics ($n = 6$) (**l**). **m** qPCR data showing expression of TPC2 and Rab7a in WT and Rab7a KO SK-MEL-5 cells ($n = 3$). Electrophysiological data (**c, f, l**) were tested for statistical significance using a one-way ANOVA test followed by Tukey's post-test. Statistical significance for **m** was determined using two-way ANOVA followed by Bonferroni multiple comparisons test and for **i** and **j** by two-tailed unpaired *t*-test. Shown are mean values ± SEM, ($n = 3$, each). *$p < 0.05$, **$p < 0.01$, ***$p < 0.001$, ****$p < 0.0001$. All $n$ numbers represent biological replicates. Source data are provided as Source Data file.

proliferation of cancer cells in vitro, and tumor growth and metastasis formation in vivo[16,19,20]. In analogy to MNT-1 melanoma cells reported previously[19], KO of TPC2 in SK-MEL-5 cells showed reduction in migration, invasion, and proliferation in all three CRISPR/Cas9 engineered TPC2 KO lines (Fig. 4a–e). Likewise, Rab7a SK-MEL-5 KO cells showed significant reduction in all three parameters: migration, invasion and proliferation (Fig. 4f, g). Rab7a protein and transcript levels were unchanged in the TPC2 KO lines (Fig. 4k). We next performed TPC2 and Rab7a knockdown (KD) experiments in several other melanoma lines. Knockdown was confirmed by either qPCR (TPC2 KD) or WB (Rab7a KD) (Fig. S5a–c). In addition to SK-MEL-5, we tested 3 lines with high Rab7a expression levels (SK-MEL-29, SK-MEL-19, UACC-62) and 3 lines with low Rab7a expression levels (SK-MEL-103, SK-MEL-147, A375). Results obtained for proliferation after TPC2 KD strongly correlated with Rab7a KD results with one exception, SK-MEL-29. All other lines showed comparable results in proliferation after either TPC2 or Rab7a KD (Fig. 5a, b). SK-MEL-5 KD

behaved like KO and was used to confirm that KD data are in accordance with KO data. SK-MEL-5, SK-MEL-19, UACC-62 and A375 showed reduced proliferation after either TPC2 or Rab7a KD while lines SK-MEL-103 and SK-MEL-147 showed no change in either KD (Fig. 5a, b). In invasion experiments, all TPC2 and Rab7a KDs showed concordant results (Fig. 5c–f). SK-MEL-5, SK-MEL-29, SK-MEL-19, and UACC-62 presented with reduced invasion after either TPC2 or Rab7a KD while lines A375, SK-MEL-103 and SK-MEL-147 showed no change in both KDs. These data suggest that TPC2 and Rab7 KD are highly conserved in their effects on proliferation and invasion in different melanoma lines.

### High MITF expressing melanoma lines show reduced proliferation and invasion after KD of either Rab7 or TPC2

MITF is a master regulator of melanocyte development with functions ranging from pigment production to differentiation and survival of melanocytes. MITF also plays a critical role in melanoma development

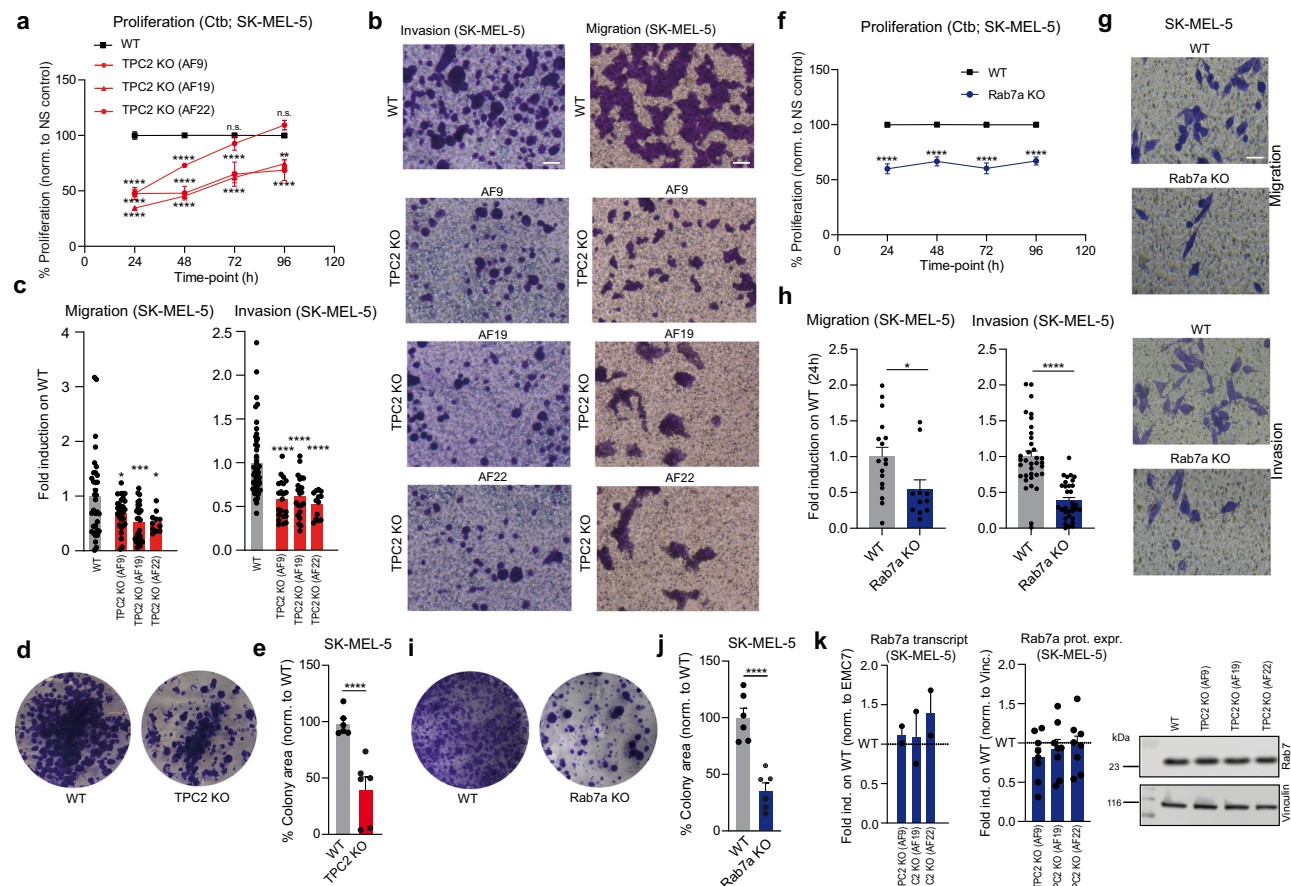

**Fig. 4 | Proliferation, migration and invasion in WT, TPC2 KO and Rab7a KO SK-MEL-5 cells.** Ctb assay assessing proliferation of SK-MEL-5 cells monitored over 96 h comparing WT cells to different clones for TPC2 KO ($n = 12$) (**a**) and Rab7a KO ($n = 18$) (**f**). Genetic ablation of either TPC2 (**b**) or Rab7a (**g**) in SK-MEL-5 melanoma line shows significantly slower invasion and migration, cells seeded on transwell chambers and monitored overnight. Statistical analysis for Boyden chamber migration and invasion experiments in TPC2 KO ($n = 11-34$) (**c**) and Rab7a KO ($n = 13-49$) (**h**) SK-MEL-5 cells. Clonogenic assay showing significant reduction in survival and growth as single colonies for both TPC2 KO (**d**) and Rab7a KO (**i**) SK-MEL-5 cells. Statistical analysis for SK-MEL-5 TPC2 KO ($n = 6$) (**e**) and Rab7a KO ($n = 6$) (**j**) plotted as colony area percentage, fold induction on WT cells. **k** qPCR ($n = 2$) and Western blot ($n = 8$) analysis indicating unchanged transcript and protein levels of Rab7a in TPC2 KO clones. Statistical significance in **a** and **f** was carried out using two-way ANOVA followed by Bonferroni multiple comparisons test ($n = 3$, each), in **c** by one-way ANOVA, and in **e**, **h**, and **j** by two-tailed unpaired $t$-test Student's $t$-test. Shown are mean values ± SEM. *$p < 0.05$, **$p < 0.01$, ***$p < 0.001$, ****$p < 0.0001$. All scale bars = 25 μm. All $n$ numbers represent biological replicates. Source data are provided as Source Data file.

and progression[34–37]. A rare functional variant of MITF[E318K] has been found to confer a 2-4-fold risk for cutaneous melanoma and may also bear risks for other cancers[38,39]. MITF, found downstream of several pathways including the canonical Wnt/GSK3β/β-Catenin pathway, contains GSK3β phosphorylation sites[30]. In the absence of Wnt signaling, GSK3β reportedly phosphorylates MITF, targeting MITF for proteasomal degradation[30]. Upon Wnt signaling, destruction complex components such as GSK3β, CK1α (Casein kinase 1α), or Axin are sequestered into LE and MVBs (multivesicular bodies), which are part of the endocytic pathway, resulting in the degradation of GSK3β and stabilization of β-Catenin and MITF[30]. While some melanoma cell lines investigated here express high MITF levels, others show low MITF expression. These data also correlate well with the Rab7a expression data (Fig. 6a–c). In human melanoma tissue samples, we likewise detected a wide range of MITF expression levels while in healthy tissue samples the levels were consistently low (Fig. 6d, e). Generally, melanoma lines, which showed reduced proliferation and invasion after either TPC2 or Rab7a KD expressed higher levels of MITF. Other lines with undetectable or very low MITF expression such as SK-MEL-103 and SK-MEL-147 showed no effect after either TPC2 or Rab7a KD on both proliferation and invasion (Fig. 5). TPC2 or Rab7 KD in A375 showed reduced invasion but not reduced proliferation while MITF

expression was likewise not detectable. In sum, the majority of melanoma lines with high MITF levels responded with a reduction in invasion and proliferation after either TPC2 or Rab7a KD/KO.

## Knockout or pharmacological inhibition of Rab7a or TPC2 result in decreased levels of MITF and increased GSK3β expression

To confirm the hypothesis that reduced destruction complex degradation and thus increased GSK3β levels go along with enhanced degradation of MITF, we performed WB experiments using the SK-MEL-5 TPC2 and Rab7a KO lines. We could confirm that MITF is reduced or absent compared to controls in both Rab7 and TPC2 KO lines, while GSK3β is increased (Fig. 6f–i). These data suggest that downregulation of TPC2 activity by Rab7a deletion results in reduced MITF, likely due to decreased endolysosomal degradation of GSK3β, the major driver of proteasomal MITF degradation. Besides increased GSK3β and decreased MITF levels, also decreased β-Catenin levels were found in both TPC2 and Rab7a KO SK-MEL-5 lines (Fig. 6f–i). β-Catenin, mutations of which are associated with many types of cancer[40] including melanoma[41], is also a key component of the Wnt signaling pathway and its degradation is controlled by GSK3β phosphorylation[41–43], further corroborating an important regulatory role of both TPC2 and Rab7a in this pathway. In addition, we could

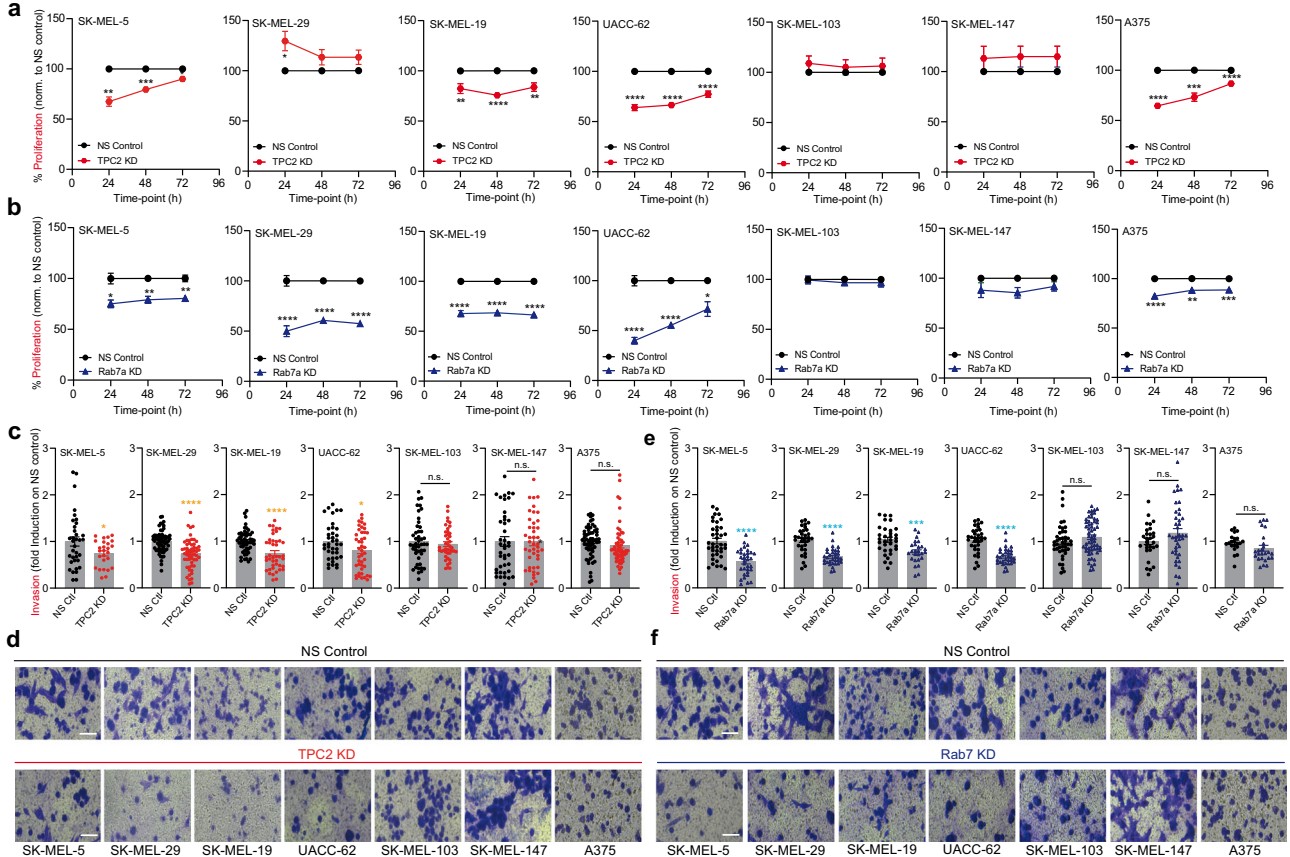

**Fig. 5 | Proliferation and invasion for different melanoma lines using knockdown siRNA.** Proliferation monitored over 72 h using Ctb assay of the following melanoma lines: SK-MEL-5, SK-MEL-29, SK-MEL-19, UACC-62, SK-MEL-103, SK-MEL-147, and A375 in non-silencing (NS) control cells compared to TPC2 KD ($n = 3$–10) (**a**) and Rab7a KD ($n = 3$–10) (**b**). Statistical analysis of cell invasiveness determined in the lines mentioned above using transwell boyden chambers coated with matrigel in both Rab7a KO ($n = 26$–64) (**c**) and TPC2 ($n = 21$–58) (**e**).

Representative images of invasion phenotype in TPC2 KD (**d**) and Rab7a KD (**f**) cells in several melanoma lines compared to NS control. Statistical significance was determined using two-way ANOVA followed by Bonferroni multiple comparisons test (**a**, **b**; $n = 3$, each) and by two-tailed unpaired $t$-test (**c**, **e**). Shown are mean values ± SEM. *$p < 0.05$, **$p < 0.01$, ***$p < 0.001$, ****$p < 0.0001$. All scale bars = 25 μm. All $n$ numbers represent biological replicates. Source data are provided as Source Data file.

show that a pharmacological inhibitor of TPC2, SG-094 likewise results in a reduction of MITF in SK-MEL-5 as well as in other melanoma lines (Fig. 6j, k). SG-094 treatment, in analogy to KO or KD of TPC2 also reduced proliferation of these melanoma lines (Fig. 6l, m).

### Rab7a acts as an enhancer of TPC2 but not vice versa

Rab7a and TPC2 KO SK-MEL-5 cells were transfected with either mCherry vector (control), Rab7a[WT]-mCherry or Rab7a[Q67L]-mCherry. Rab7a[Q67L] rescued proliferation and Rab7a[WT] and Rab7a[Q67L] both rescued invasion defects in Rab7a KO SK-MEL-5 cells (Fig. 7a–l). Of note, the TPC2[M484L] GOF variant or TPC2[WT] in combination with TPC2-A1-P agonist, or TPC2-A1-P agonist alone also rescued proliferation and invasion in Rab7a KO SK-MEL-5 cells. By contrast, in TPC2 KO SK-MEL-5 cells Rab7a[WT] or Rab7a[Q67L] had no or much reduced rescue effects compared to their effects in Rab7a KO SK-MEL-5 cells, for both proliferation and invasion, suggesting that without TPC2 Rab7a cannot sufficiently rescue. TPC2 KO was rescued by TPC2[M484L], TPC2[WT], or TPC2[WT] in combination with TPC2-A1-P but not with TPC2-A1-P alone (TPC2-A1-P is expected not to work in TPC2 KO, confirming specificity; Fig. 7i–l). These data show that Rab7a and TPC2 depend on each other. However, while loss of Rab7a can be efficiently rescued with TPC2 overexpression (OE), loss of TPC2 can either not or only marginally be rescued by Rab7a OE. In line with this, increased GSK3β levels in Rab7a

KO could be reduced again when overexpressing either Rab7a[WT] or Rab7a[Q67L] (Fig. S6a). Vice versa however, increased GSK3β levels in TPC2 KO could not be reversed by overexpressing either Rab7a[WT] or Rab7a[Q67L] (Fig. S6b). Of note, overexpression of GSK3β was able to further reduce proliferation in both Rab7a and TPC2 KO SK-MEL-5 lines (Fig. S6c). Similar experiments have been performed for β-Catenin. Thus, overexpressing Rab7a[Q67L] could rescue the effects on β-Catenin in Rab7 KO but not in TPC2 KO (Fig. S6d, e). Importantly, over-expression of MITF was able to increase proliferation in WT, confirming a direct effect of MITF on melanoma cell proliferation in SK-MEL-5, while both Rab7a and TPC2 KO lines did not show a difference in proliferation compared to empty vector expression after MITF OE, likely because MITF OE in the KOs can still not keep up with the continued degradation (Fig. S6d–h). In sum, these findings corroborate the hypothesis that Rab7a acts as an enhancer or "effector" of TPC2 but not vice versa, with "effector" being defined here as an upstream molecule (Rab7a) interacting with a downstream target (TPC2) to increase its activity.

### Melanoma growth formation and dissemination in vivo is mediated by Rab7a acting as an enhancer of TPC2 activity

An ectopic tumor model with murine melanoma B16F10-luc cells was chosen for in vivo experiments to investigate tumor growth and

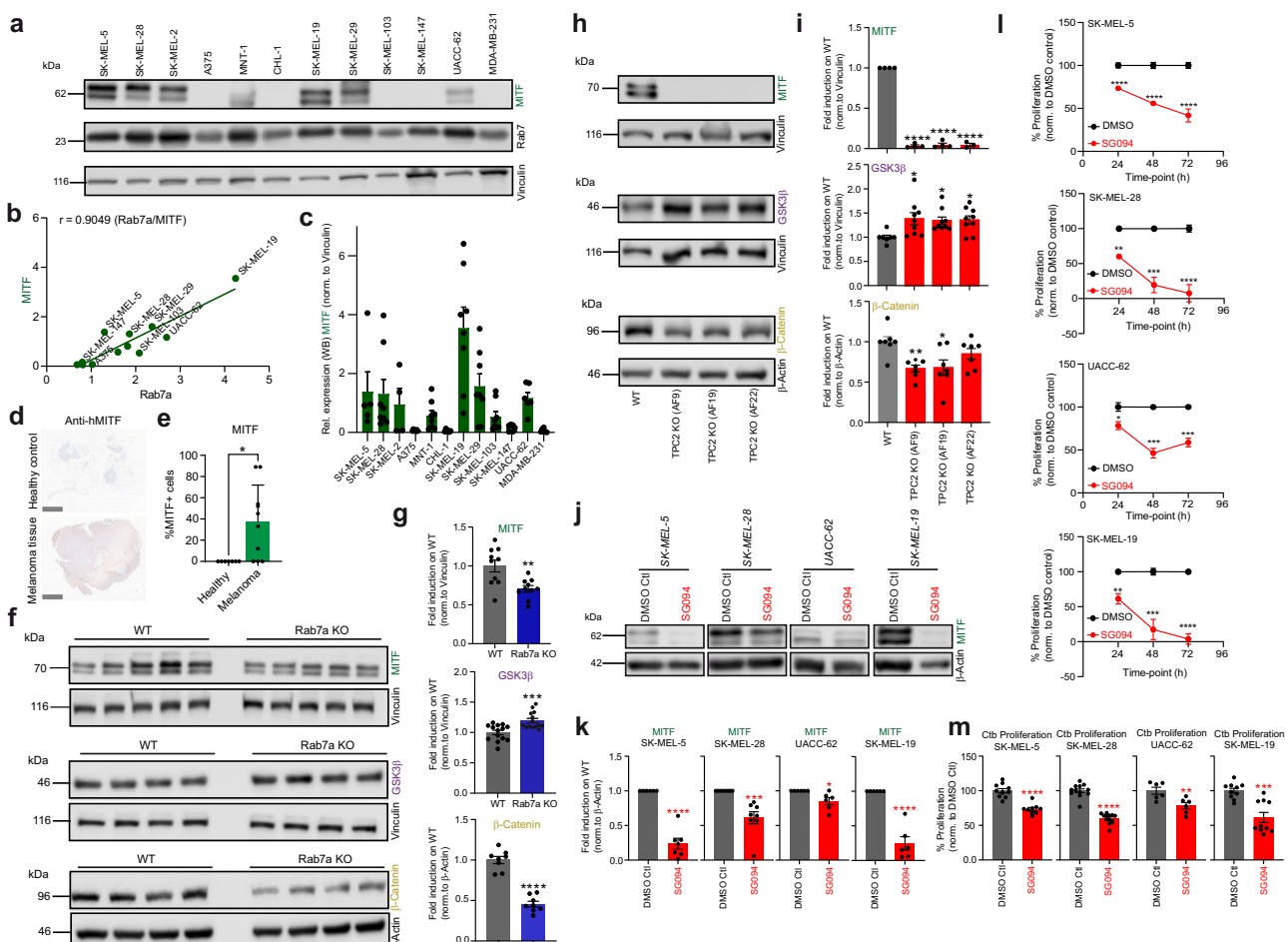

**Fig. 6 | Expression of MITF and GSK3β in different melanoma lines and effects of Rab7a or TPC2 KO or small molecule blockers. a** Representative Western blots for MITF and Rab7 protein expression in different melanoma lines and in the breast cancer line MDA-MB-231, normalized to Vinculin. **b**, **c** Correlation plot for MITF/Rab7 expression (**b**) and statistical analysis of experiments as shown in (**a**) (mean values ± SEM, $n = 5$–8). Data points represent biological replicates (**c**). **d** Representative images of sections from healthy lymphnode (male, abdomen) and melanoma lymphnode metastasis (male, iliacal) samples stained with hMITF antibody (IHC). Scale bars = 5 mm. **e** IHC evaluation was carried out considering the percentage of stained tumor cells. Statistical significance was assessed by two-tailed unpaired $t$-test, $*p = 0.0125$ (mean ± SD). One dot corresponds to one independent human donor ($n = 10$ for each condition). Genetic knockout of either Rab7a (**f**, **g**) or TPC2 (**h**, **i**) in SK-MEL-5 cells shows reduction in the protein levels of

MITF and β-Catenin but increased expression of GSK3β. Statistical analysis for the expression levels of MITF, GSK3β, and β-Catenin, WT vs. Rab7a KO or TPC2 KO is shown in (**g**, **i**), respectively; significance determined by two-tailed unpaired $t$-test (**g**) or by one-way ANOVA (**i**). Shown are mean values ± SEM, $n = 8$–14 in (**g**) and 3–9 in (**i**). $*p < 0.05$, $**p < 0.01$, $***p < 0.001$, $****p < 0.0001$. **j**, **k** Representative blots for MITF protein expression in different melanoma lines after treatment with the TPC2 inhibitor SG-094 (7 μM) or DMSO control for 24 h. Statistical significance was determined by one-way ANOVA. Shown are mean values ± SEM, $n = 5$–7. $*p < 0.05$, $***p < 0.001$, $****p < 0.0001$. Proliferation experiments showing effect of SG094 treatment (7 μM) in SK-MEL-5 and other SK-MEL melanoma lines, significance determined by one-way ANOVA (**l**) or two-tailed unpaired $t$-test (**m**). Shown are mean values ± SEM. $*p < 0.05$, $**p < 0.01$, $***p < 0.001$, $****p < 0.0001$. All $n$ numbers represent biological replicates. Source data are provided as Source Data file.

dissemination. After confirming expression of TPC2 and Rab7a (Fig. S7a), we generated murine TPC2 and Rab7a CRISPR/Cas9 KOs in B16F10-luc cells (Fig. S7b, c). We validated these cells by qPCR and WB analysis to assess successful KO of TPC2 and Rab7a, respectively (Fig. S7d–g). In vitro we further characterized these cells by using a wound healing assay. In this assay both Rab7 and TPC2 KO showed significantly hampered wound healing and migration (Fig. S7h, i). C57BL/6BrdCrHsd-Tyr<sup>c</sup> mice were injected (subcutaneous injections into the flank) with either WT, TPC2 KO or Rab7a KO B16F10-luc cells. Mice injected with either TPC2 KO or Rab7a KO B16F10-luc cells exhibited significantly decreased tumor weights and reduced dissemination as well as reduced MITF and β-Catenin staining in ex vivo tumor tissue (Figs. 7m–o and S7j–m) compared to mice injected with WT B16F10-luc cells. These findings are consistent with our in vitro results. Furthermore, the reduction in tumor weight observed with Rab7a KO cell injections could be partially reversed by treatment with

the TPC2-A1-P agonist, with tumor weights reaching levels that were not significantly different from WT (day 14). This effect became already apparent on day 7, as indicated by bioluminescence data, where treatment with TPC2-A1-P significantly reversed the Rab7a KO phenotype based on bioluminescence intensity. In contrast, ML-098, a Rab7 small molecule agonist was not able to reverse the TPC2 KO effect on tumor weight (Fig. 7m–o), further supporting our in vitro observations. Taken together these data corroborate our hypothesis that Rab7a KO and TPC2 KO exert similar effects on cancer hallmarks and that the effect of Rab7a is mediated through TPC2, as the loss of Rab7a can be largely compensated with TPC2 agonist or overexpression but not vice versa.

## Discussion
We have shown here that knockout of either Rab7a or TPC2 in melanoma lines, in particular those that express high levels of MITF result in

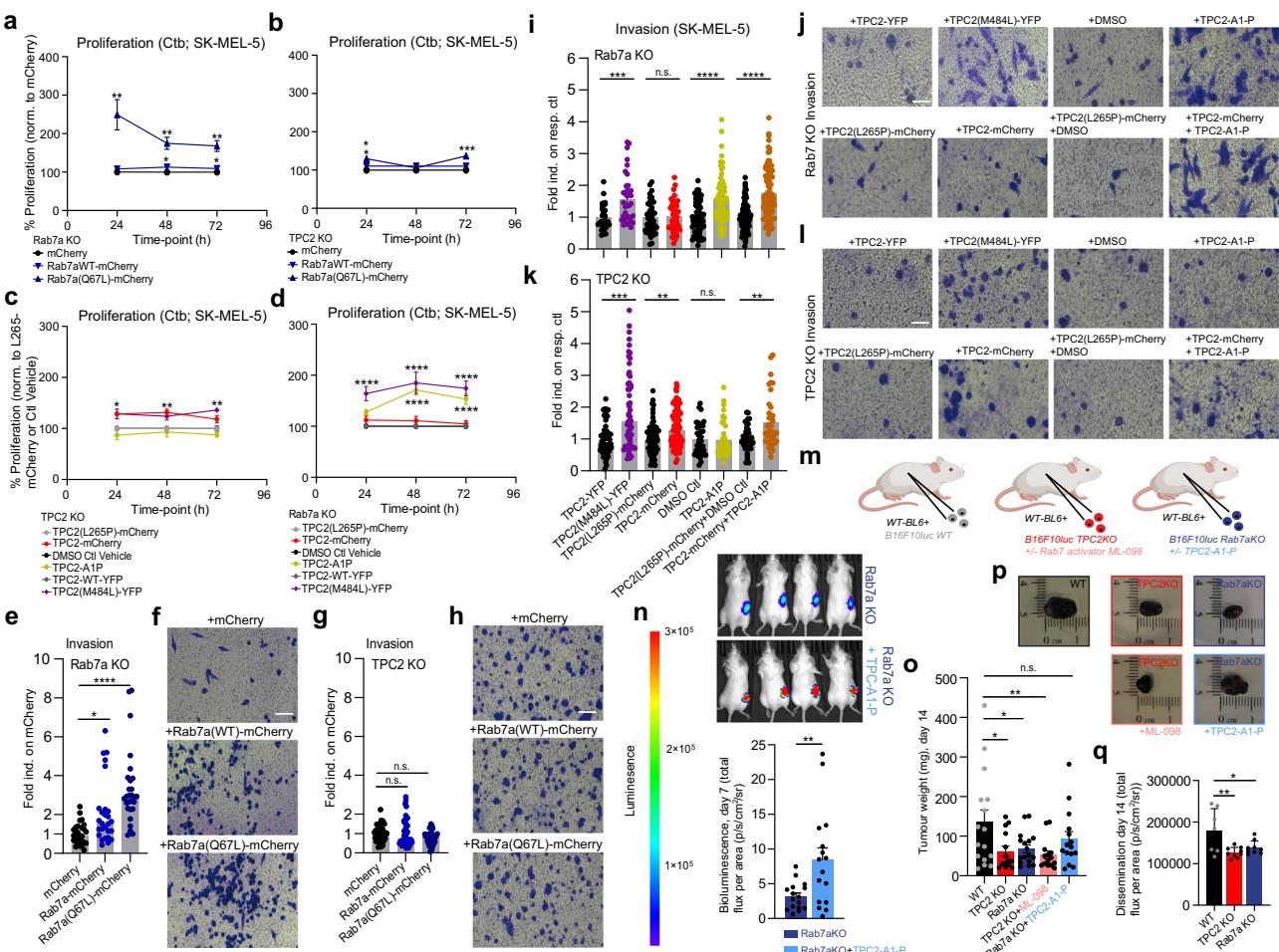

**Fig. 7 | Rescue experiments in SK-MEL-5 TPC2 and Rab7a KO.** Proliferation of Rab7a KO (**a**) or TPC2 KO (**b**) cells expressing vector alone, Rab7^WT- or Rab7^Q67L-mCherry, assessed for 24-, 48-, and 72-h and normalized to vector. Proliferation assessed for 24-, 48-, and 72-h for Rab7a KO (**c**) or TPC2 KO (**d**) SK-MEL-5 cells expressing TPC2^WT-mCherry normalized to TPC2^L265P-mCherry, TPC2^M484L-YFP normalized to TPC2^WT-YFP, and treatment with the agonist TPC2-A1P normalized to DMSO control. **e–h** Representative images of the invasive phenotype determined by OE of mCherry vector, Rab7^WT- or Rab7^Q67L-mCherry in Rab7a KO and TPC2 KO, and statistical analysis shown in (**e**) and (**g**), respectively. Statistical significance was determined using one-way ANOVA, mean values ± SEM. **i–l** Representative images and statistical analysis of the invasive phenotype determined by OE of TPC2^WT-mCherry, TPC2^M484L-YFP, and/or treatment with TPC2-A1P in Rab7a and TPC2 KO, normalized to resp. control. Statistical significance in **a–e**, **g**, **i**, **k** was determined using one-way ANOVA followed by Bonferroni multiple comparisons test, mean values ± SEM (**a–d**: n = 9–15; **e**, **g**: n = 4; **i**: n = 3–9, **k**: n = 5–8). *p < 0.05, **p < 0.01, ***p < 0.001, ****p < 0.0001. **m–p** In vivo experiments showing tumor growth/

weight after subcutaneous injection of B16F10luc WT, TPC2 KO or Rab7a KO cells into 5–6 week old C57Bl/6-Tyr mice. Treatment with vehicle control, ML-098 (0.04 mg/g) or TPC2-A1-P (0.02 mg/g) was performed daily (**m**, "Created in BioRender". BioRender.com/n88t710."). Bioluminescence images (d7 after injection) and bioluminescence signal intensities of tumors, mean ± SEM (n = 15 for Rab7a KO + TPC2-A1-P, n = 17 for Rab7a KO). Statistical significance was assessed by two-tailed unpaired t-test, **p < 0.0001 (mean ± SEM) (**n**). Tumor weights and representative tumors at the endpoint (day 14) are shown in (**o**, **p**). Statistical significance was determined using one-way ANOVA, mean values ± SEM (WT n = 16, TPC2 KO ± ML-098 n = 14 each, Rab7a KO n = 17, Rab7a KO + TPC2-A1-P n = 15), *p < 0.05, **p < 0.01. **q** In vivo experiments showing tumor cell dissemination after intravenous injection of 2 × 10^5 B16F10-luc WT, TPC2 KO or Rab7a KO cells. Data from day 14 (endpoint) are shown. Statistical significance was determined using ordinary one-way ANOVA, mean ± SD (WT n = 7, TPC2 KO and Rab7a KO n = 9), *p < 0.05, **p < 0.01. Source data are provided as Source Data file.

similar phenotypes i.e., reduction of proliferation, migration, invasion, and tumor growth in vitro and in vivo. We further show that Rab7a, which physically and functionally interacts with TPC2, is a strong enhancer of TPC2 activity. Whether this is a direct interaction or an interaction mediated by one or more additional proteins forming a complex, remains to be further explored. Importantly, in endolysosomal patch-clamp and GCaMP based Ca²⁺ imaging experiments activation of TPC2 is (independent of the respective ligand used) several fold increased when Rab7a is present. We further found that KO of Rab7a reduces endogenous TPC2 activity in SK-MEL-5 melanoma cells, corroborating the findings in OE HEK293 cells. On the other hand, the activity of the lysosomal cation channel TRPML1 was Rab7a-independent since coexpression with Rab7a showed no difference in TRPML1 activation in endolysosomal patch-clamp experiments after

application of either a TRPML1 synthetic small molecule agonist, ML1-SA1 or the endogenous agonist PI(3,5)P₂. In line with these results, activity of TRPML1 in SK-MEL-5 melanoma cells was not altered by the KO of Rab7a. Our data imply that Rab7a acts as an enhancer of TPC2 activity to regulate melanoma proliferation, migration, invasion, and tumor growth. Effects on proliferation and invasion in Rab7a KO melanoma cells could be rescued by TPC2 OE and/or TPC2 activation but vice versa TPC2 KO could either not be rescued by Rab7a OE (invasion) or with much reduced rescue efficacy (proliferation). In in vivo experiments in mice, we could show that dissemination and tumor growth/weight were reduced in mice injected with TPC2 KO or Rab7a KO B16F10-luc melanoma cells as compared to WT B16F10-luc melanoma cell injection. Additionally, reduced tumor growth/weight in mice injected with Rab7a KO B16F10-luc melanoma cells could be

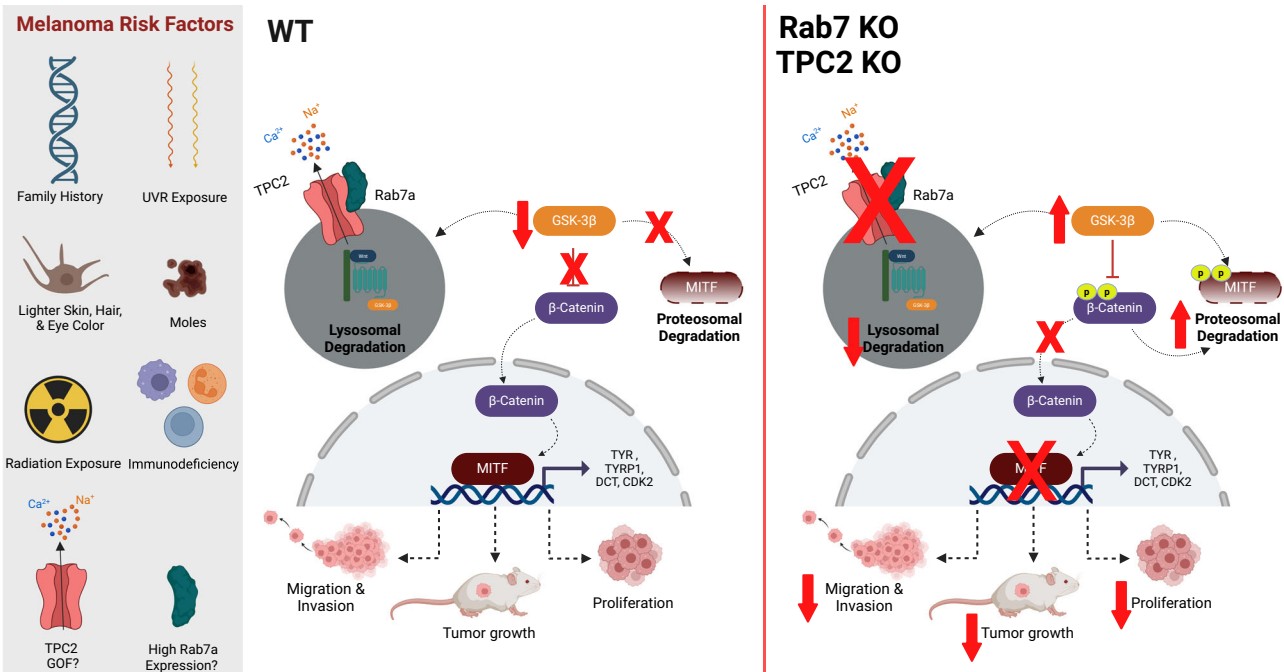

**Fig. 8 | Proposed mechanism of β-Catenin and MITF regulation in SK-MEL-5 WT and Rab7/TPC2 KO cells.** In WT cells, Rab7a upregulates activity of TPC2 which induces endolysosomal degradation of the GSK3β complex. Downregulation of GSK3β leads to decreased phosphorylation of β-Catenin, which then translocates in its dephosphorylated state to the nucleus and activates MITF transcription. The transcription factor MITF induces the transcription of various genes upregulating cell migration, invasion, proliferation and tumor growth. Furthermore, decreased GSK3β levels lead to decreased phosphorylation of MITF, hence less proteosomal degradation of MITF. When Rab7a or TPC2 is dysfunctional or knocked out, GSK3β phosphorylates β-Catenin and MITF promoting their proteosomal degradation, and blocking β-Catenin from translocation to the nucleus. Less MITF transcription leads to a decrease in migration, invasion, proliferation and tumor growth, suggesting that TPC2 GOF SNPs and high Rab7a expression may be risk factors for melanoma. "Created in BioRender. BioRender.com/c52w590".

partially reversed by treatment with TPC2-A1-P to a level, which was not significantly different from WT control. By contrast, a Rab7 agonist, ML-098 was not able to significantly reverse tumor growth in the TPC2 KO condition.

Effects of TPC2 or Rab7a KO or KD on proliferation, migration, and invasion were seen particularly in melanoma lines expressing high levels of MITF, and MITF levels were strongly depleted or reduced after either KO or KD of TPC2 or Rab7a. MITF proteasomal degradation is reportedly enhanced when the endolysosomal degradation of GSK3β is reduced[19,30]. Based on our data, including data confirming reduced endolysosomal function in Rab7a as well as TPC2 KO SK-MEL-5 melanoma cells (Fig. S8), as reported previously for other cells[1-4,13-18], we postulate that activation of TPC2, enhanced by Rab7a promotes endolysosomal activity and thus degradation of GSK3β. This ultimately results in less GSK3β being available for MITF degradation. Vice versa, loss of either TPC2 or Rab7a reduces endolysosomal degradative activity, leaving more GSK3β available for the proteasomal degradation of MITF. As further evidence that components of the GSK3β/MITF axis are affected by Rab7a/TPC2 activity, we demonstrated that β-Catenin is reduced in Rab7a as well as in TPC2 KO melanoma lines. β-Catenin is directly dependent on GSK3β phosphorylation for its degradation, and it is a known regulator of melanoma cell growth, with MITF as a critical downstream target[44,45] (Fig. 8). Furthermore, CK1α, which is like GSK3β an important component of the destruction complex was found to be significantly increased in both Rab7a and TPC2 KO (Fig. S9a). In addition, we also examined other pathways with relevance for GSK3β/MITF modulation such as the Akt or CREB pathways[46] (Fig. S9b, c). While pCREB/CREB ratios were not significantly different between KOs and WT, pAkt/Akt ratios were increased in the KOs, which reportedly should result in increased MITF levels[46]. However, since in both Rab7a and TPC2 KO lines we found reduced MITF levels, the increase in Akt may likely reflect a compensatory effect in response to the changes in the GSK3β/β-Catenin/MITF axis due to TPC2 or Rab7a KO. Of note, the ratios of pAkt/Akt showed very high variability (Fig. S9b). As additional evidence for an enhanced degradation of β-Catenin, we confirmed, in WB experiments increased phosphorylated β-Catenin levels in TPC2 KO cells compared to WT (Fig. S9d, e). Of note, changes in β-Catenin dependent (WNT3A driven) as well as β-Catenin independent (WNT5A driven) Wnt signaling (canonical versus non-canonical) have been described in the context of cancer and melanoma. The former one drives proliferation, the latter one reportedly enhances the invasiveness of cancer and melanoma cells. However, this distinction seems less rigid as previously thought as concomitant crosstalk and mutual regulation were found to be common[46,47]. Canonical Wnt signaling results in reduced GSK3β levels and thus less phosphorylated β-Catenin destined for degradation, thus leading to enhanced MITF expression. KO of TPC2 or Rab7 result in increased GSK3β levels and reduced β-Catenin, and ultimately reduced or absent MITF expression, likely explaining the observed reduced proliferative activity of these cells. While canonical Wnt signaling initiates β-Catenin dependent signaling, thus increasing MITF expression, and Rab7/TPC2 KOs negatively interfere with this, non-canonical Wnt signaling inhibits, besides having multiple other effects β-Catenin signaling (via SIAH2), thus reducing MITF expression, but in a GSK3β independent manner[46]. Several reports claim that MITF follows a rheostat model[48] according to which MITF can result in varied cellular responses in melanoma cells based on its activity. In brief, at peak MITF levels, melanoma cells express differentiation genes, promoting a pigmented phenotype and terminal differentiation. At an intermediate MITF level cells are in a reversible proliferative state, suppressing differentiation. At lower level MITF promotes invasiveness and cells exhibit more stem-cell-like properties but low proliferative and pigmentation capacities. At its lowest level, MITF drives senescence and apoptosis[49,50]. This model has

been discussed controversially in recent years[49–52] e.g., by Haass et al.[51], and Wellbrock and Arozarena[50] pointed out that the regulation of MITF expression and function is much more dynamic and versatile than previously thought, suggesting that the rheostat model might be too restrictive. We show here that KO, KD, or pharmacological inhibition of Rab7a or TPC2 results not only in reduced proliferation, but also in reduced invasion and migration, most consistently in high MITF expressing melanoma lines, which after TPC2 or Rab7a KO or KD, or TPC2 inhibitor treatment (SG-094) show much reduced or absent MITF expression, in line with the reduction in β-Catenin levels found for the KOs in vitro and in vivo. Several works suggest that β-Catenin may not only promote cancer and melanoma cell proliferation but also migration and invasion, albeit the latter remains controversial[41,53–55]. Furthermore, in vivo studies in a mouse model of melanoma have shown that β-Catenin is a central mediator of melanoma metastasis to the lymph nodes and lungs. Nevertheless, the role of β-Catenin and MITF in cancer and melanoma remains incompletely understood. The surprising finding that an endolysosomal cation channel, TPC2 is involved in the regulation of GSK3β, and consequently β-Catenin and MITF levels in concert with Rab7a as an enhancer of the activity of TPC2 may add an additional layer of complexity, but may also contribute to a better understanding of the regulation of these central signaling components in cancer and melanoma and thus for novel therapeutic interventions[46,47,53,56].

In sum, our in vitro KD and KO data as well as our rescue studies and in vivo experiments suggest a correlation between Rab7a and TPC2 activity, with Rab7a acting as an enhancer of TPC2 activity, promoting tumor hallmarks such as proliferation, migration, invasion, and tumor growth, in particular in melanoma cells expressing high levels of MITF[9]. In accordance with our findings, Alonso-Curbelo et al. (2014) postulated that Rab7 is an early-induced melanoma driver and found reduced proliferation after Rab7 shRNA treatment for different melanoma lines e.g., UACC-62 or WM-164 while Rab7 shRNA treatment was much less efficient in reducing proliferation of SK-MEL-103 cells, becoming significant in the latter only after 4 days. Instead, after 3 days the effect was not significant, in line with our results[9]. Likewise, non-melanoma cancer lines were barely affected by Rab7 shRNA treatment. Alonso-Curbelo et al. (2014) further found reduced tumor volume after Rab7 shRNA treatment, again with much more prominent effects in UACC-62 (high MITF) as compared to SK-MEL-103 or SK-MEL-147 (low MITF)[9]. Our data provide a new concept of how a small GTPase, Rab7a affects cancer/melanoma hallmarks by controlling the activity of the endolysosomal cation channel TPC2 with effects on GSK3β degradation and thus β-Catenin and MITF levels.

## Methods

### Plasmids
The human TPC2-YFP (C-terminally tagged) plasmid[26] and the TRPML1-YFP plasmid[57,58] were used for all patch-clamp measurements as well as human Rab7a-mCherry (N-terminally-tagged). All plasmids used for FRET measurements were generated using restriction-insertion cloning. For the construct mTq2-Rab7a, the cDNA sequence of mTurquoise was fused 5′ to the sequence of human Rab7a (CCDS3052.1) and cloned into a pcDNA3.1+ expression vector. The mTq2-Rab7b (CCDS73011.1) construct and the mTq2-Rab5 (CCDS2633.1) construct were generated by replacing the Rab7a in mTq2-Rab7a. mTq2-Rab7a$^{Q67L}$ and mTq2-Rab7A$^{T22N}$ were generated by editing the mTq2-Rab7a via Quikchange (Agilent). Human TPC1 (CCDS31908.1) and TPC2 (CCDS8189.1) were cloned into a pcDNA3.1+ expression vector together with a C-terminal mVenus. Single mTurquoise2 and mVenus were each cloned into pcDNA3.1+. Three tandem-constructs containing both mTurquoise2 and mVenus separated by linker sequences of different lengths were generated. For the shortest tandem-construct (Dimer-3AA) both fluorophores are separated by a GSG-linker. The second tandem-construct (Dimer-42AA) contains a 42-amino acid-long randomized sequence separating both fluorophores.

The third tandem-construct (Dimer-2A) contains a 2A-peptide as a linker sequence. The original plasmid was a kind gift from Prof. Dorus Gadella (Addgene #98885). mNeongreen was exchanged with mVenus. The following plasmids were used: hTRPML1-YFP plasmid[58], the hTPC2-YFP[26] and hTPC2$^{M484L}$-YFP[26], hTPC2-GCaMP6s[31], hTPC2$^{L265P}$-GCaMP6s[31], hTPC2-mCherry[31], and hTPC2$^{L265P}$-mCherry[31]. The hTPC2-myc plasmid was a gift from Prof. Sandip Patel. For hRab7a$^{WT}$-mCherry: Rab7A was amplified using High Fidelity PCR Kit (Roche). Sequences were as follows: fw: 5′-CTGTACAAGGGATCCGGAATGACCTCTAGGAAGAAAGTGTTGCTGAAG-3′ and rev: 5′-GTCGGGCCCTCAGCAACTGCAGCTTTCTGCC-3′ and cloned into pcDNA3.1+ using BamHI (Thermo Fisher) and ApaI (Thermo Fisher) via restriction and ligation cloning (Thermo Fisher). The following steps apply to all constructs: The plasmids are transformed into DH5-a (Thermo Fisher). Bacteria were plated on LB agar plates containing 100 μg/ml Ampicillin. Single colonies were picked and cultured overnight in 100 μg/ml Ampicillin LB liquid media at 37 °C at 135 RPM. Plasmid purification was conducted with (Macherey Nagel). mCherry was amplified using High Fidelity PCR Kit (Roche). Sequences were as follows: fw: 5′-GGAGACTCGGCTAGCATGGTGAGCAAGGGCGAGG-3′ and rev: 5′GGGTCCGGATCCCTTG GTACAGCTCGTCCATGCC-3′ and cloned into the Rab7 containing vector N-terminally using NheI (Thermo Fisher) and BamHI (Thermo Fisher) as described above. For hRab7a$^{Q67L}$-mCherry: hRab7a$^{WT}$-mCherry was altered via Quikchange (Agilent). Sequences as follows: 5′-GGACACAGCAGGACTGGAACGGTTCCAGT-3′ and rev: 5′-ACTGGAACCGTT CCAGTCCTGCTGTGTCC-3′. The entire insert (mCherry-Rab7A[Q67L]) was subcloned into pcDNA3.1 using NheI (Thermo Fisher) and ApaI (Thermo Fisher). For hRab7a$^{T22N}$-mCherry: hRab7a$^{WT}$-mCherry was altered using a site-directed mutagenesis Kit (Agilent). Sequences as follows: fw: 5′-GGAGATTCTGGAGTCGGGAAGAACTCACTCATGAACCAG-3′ and rev: 5′-CTG GTTCATGAGTGAGTTCTTCCCGACTCCAGAATCTCC-3′. The entire insert (mCherry-Rab7A[T 22N]) was subcloned into pcDNA3.1 using NheI (Thermo Fisher) and ApaI (Thermo Fisher).

### Two-hybrid FRET experiments
For FRET experiments 17 mm glass-bottom imaging dishes (Ibidi 81218-200) were treated for 2 h with Poly-L-Lysine. Cells were seeded two days prior to measurement. Cells were transfected with Lipofectamine 2000 (Thermo Fisher) one day before measurement. Each FRET pair was transfected using 0.7 μg of donor plasmid DNA (mTq2 constructs) and 1.5 μg of acceptor plasmid DNA (mVenus constructs) to compensate for the lower expression rate of the channel-proteins (TPC1, TPC2, TRPML1). Images of live HEK293 cells were acquired using a Zeiss LSM980 confocal microscope using a 60X oil Objective at excitation wavelengths of 445 nm and 515 nm for mTurquoise2 and mVenus, respectively. Images were acquired in three emission channels using bidirectional line-scans: mTq2DIRECT (445 nm ex; 455-526 nm em), mTq2FRET (445 nm ex; 455-526 em), mVenusDIRECT (515 ex; 526-561 em). Images were analyzed using FIJI and processed with a weak Gaussian-blur. Regions of interest were drawn around mVenus-positive lysosomes. Using a FIJI macro, the mean intensity of fluorescence in each ROI in all channels were calculated and formatted in a.csv table. Calibration constants were obtained from cells expressing single mTurquoise2, single mVenus and tandem construct, respectively. FRET Two-Hybrid curves were generated using a custom MatLab function (MathWorks). Calculation was performed according to Liu et al.[59].

### siRNA knockdown and compounds
Cells were silenced using small interfering RNA (siRNA) and transfection in the melanoma cell lines using lipofectamine 3000 regent (Thermo Fisher). The following siRNAs were used for experiments: hRab7a (Sigma-Aldrich), hTPC2 (Dharmacon SMARTpool) in a mixture of four oligonucleotide duplexes and non-targeting Scramble control siRNA (Dharmacon SMARTpool) was used as a control. For rescue experiments and whole-endolysosomal patch-clamp recordings, the following

small-molecules/drugs were used: PI(3,5)P$_2$-di8 (Echelon Biosciences), ATP-Mg (Sigma-Aldrich), GTP (Sigma-Aldrich), CID1067700 (Sigma-Aldrich), TPC2-A1-P[31] and TPC2-A1-N[31], and SG094[16].

## Cell culture

Human embryonic kidney HEK293 cells were cultured in DMEM (Gibco, containing 1 g/L glucose) supplemented with 100U/ml of Penicillin-streptomycin (Pen/Strep) (Sigma-Aldrich), and 10% of fetal bovine serum (FBS) (Thermo Fisher). The following cancer lines were cultured in DMEM (4.5 g/L glucose, Thermo Fisher) supplemented with 100U/ml of Penicillin-streptomycin and 10% of FBS: SK-MEL-5, SK-MEL-28, A375, SK-MEL-19, SK-MEL-29, SK-MEL-103, SK-MEL-147, UACC-62, CHL-1, Huh-7, HepG2, MDA-MB-231, MCF-7, SK-BR-3, SKOV3, Hela, and Caco-2. MNT-1 cells were cultured in MEM (Thermo Fisher), 20% FBS, 10% AIM-V (Thermo Fisher), 1% sodium pyruvate (Thermo Fisher), and 1% Pen/Strep. The neuroblastoma cell line, U87MG, was cultured in MEM eagle with 10% FBS and 1% Pen/Strep. While cell lines: Panc-1, SK-MEL-2 and B16F10luc were cultured in RPMI 1640 (Thermo Fisher), 10% FBS, and 1% Pen/Strep. Cells were washed with 1x phosphate buffered saline (PBS) (Thermo Fisher) and detached from the surface of cell culture plates using Trypsin-EDTA (1x) (Thermo Fisher). Cell lines were maintained in an incubator at 37 °C with 5% carbon dioxide.

## CRISPR/Cas9 knockout lines

TPC2 knockouts were generated in the SK-MEL-5 melanoma cell line by targeting exon 3 of the TPCN2 gene, using guide RNAs designed for introns 2/3 and 3/4 [34] (Fig. S4). In total, three SK-MEL-5 TPC2 KO clones (AF9, AF19, and AF22) were used for the experiments in this study. Human and mouse Rab7a KOs were generated by targeting Exon 2. Guide RNAs were designed in Intron 1/2 and Intron 2/3. Murine TPC2 KO strategy in the B16F10-luc cells was adapted from Müller et al.[16]. Cloning of sgRNAs was performed into eSpCas9(BB)_2A_GFP (PX458), subsequently selection and single cell sorting was performed via FACS (BD FACSAria Fusion). All KO clones were validated using genotyping analysis by genomic PCR, genomic sequencing, and RT-qPCR. Moreover, hTPC2 KO clones were further validated using endolysosomal patch clamp experiments, measuring lysosomal currents using agonist stimulation (TPC2-A1P) and ATP as a blocker. The hRab7a KO clone (C1x17) in SK-MEL-5 and the mRab7a KO in B16F10-luc were validated using western blotting. The following antibodies were used for human Rab7 protein from Cell Signaling Technology: 2094S and 9367S, binding around residues Asp193 and Glu188, respectively, corresponding to Exon 6.

## Whole-endolysosome manual patch-clamp

For endolysosomal patch-clamp recordings, HEK293 and SK-MEL-5 cells were seeded into 24 well plates with poly-L-lysine (Serva) coated coverslips with a cell density of 60-70%, followed by transient transfection of the cells with proteins of interest using TurboFect Transfection Reagent (Thermo Fisher) in the case of HEK cells. SK-MEL-5 cells were used for endogenous assessment of endolysosomal currents. Co-transfection of TPC2 and Rab7a, and TRPML1 and Rab7a plasmids was performed in a ratio of 2:1, respectively. After 12-24 h transfection, HEK293 cells were treated overnight with 1 mM apilimod (Axon Medchem) to enlarge lysosomes and late endosomes. The compound was washed out before patch-clamp experiments were performed. Data were digitized at 49 kHz and filtered at 2.8 kHz. All the currents measured were recorded using an EPC-10 patch-clamp amplifier and PatchMaster acquisition software (HEKA). Recording pipettes were polished with a 4-6 MΩ resistance. Liquid junction potential was corrected. All experiments were conducted at room temperature (23-25 °C). The cytoplasmic solutions were replaced after application of agonists or antagonists. Individual ramp current recordings were extracted at the -100mV current amplitudes. Unless otherwise stated, the extracellular/bath solution consisted of 140mM K-MSA, 5 mM KOH, 4 mM NaCl, 0.39 mM CaCl$_2$, 1 mM EGTA, 10 mM HEPES with a 7.2 pH adjusted with KOH (300 mOsm adjusted with D-(+)-glucose). The pipette/luminal solution contained 140 mM Na-MSA, 5mM K-MSA, 2 mM Ca-MSA, 1 mM CaCl$_2$, 10 mM HEPES, 10 mM MES adjusted with methanesulfonic acid (310 mOsm adjusted with D-(+)-glucose) to pH 4.6. In all of the experiments, 500-ms voltage ramps from -100 to +100 mV were applied every 5 s. All statistical analysis was done using Origin8 or GraphPadPrism software.

## Ca$^{2+}$ imaging

Ca$^{2+}$ imaging was performed using an inverted Leica DMi8 live cell microscope. Recordings and adjustments were executed within LAS X software. At first, the DMEM was washed away from the 6-well plates with HEK293 cells, transfected with GCaMP6-tagged plasmids. Ca$^{2+}$-free buffer was used to carefully rinse the wells with cells before placing the glass coverslips to an imaging chamber. All GCaMP6 experiments were conducted in Ca$^{2+}$-free buffer comprising 138 mM NaCl, 6 mM KCl, 1 mM MgCl$_2$, 10 mM HEPES, and 5.5 mM D-glucose monohydrate (adjusted to pH = 7.4 with NaOH). To create Ca$^{2+}$-free environment around the cells, 450 µL of Ca$^{2+}$-free buffer was added to the chamber slowly, not to wash away the cells. The osmolarity of the Ca$^{2+}$-free buffer was also 300 mOsmol/L. GCaMP6 was excited at 470 nm (GFP excitation wavelength) and emitted fluorescence was captured with a 515 nm long-pass filter. Images were obtained every 2.671 sec with 63x objective. When the GCAMP6-tagged plasmid was co-transfected with mCherry-tagged plasmids the co-localization of GCaMP6 and mCherry signals was visually verified and these cells were chosen for measurement. mCherry was excited at 568 nm and the emitted fluorescence was captured at a 590 nm filter. For quantification of change in Ca$^{2+}$ acquired by GCaMP6 fluorescence, regions of interest ROIs were drawn around each cell, expressing only GCaMP6 or only co-localized GCaMP6 and mCherry. The background area without cells was selected for manual background subtraction. Fluorescence was calculated using LAS X software. The baseline value (F$_0$) was acquired by averaging fluorescence from a 30 sec recording before the addition of a compound. Change in GCaMP6 fluorescence (ΔF) was normalized to the baseline value (ΔF/F$_0$) for the data presentation.

## RT-qPCR

Total RNA was extracted from cell lines using RNeasy Plus Mini Kit (Qiagen) according to the manufacturer's protocol. cDNA was synthesized from total RNA with RevertAid First Strand cDNA Synthesis Kit (Thermo Scientific). Real-time quantitative Reverse Transcription PCR (qPCR) was performed using LightCycler 480 SYBR Green I Master Mix (Roche) and Light Cycler 480 Instrument (Roche, Light Cycler 480 software v1.5.1). Reactions were carried out in triplicates under conditions according to manufacturer's recommendations. The sequences for the primers used for RT-qPCR were: hHPRT qPCR: fw: 5'-TGGCGTCGTGATTAGTGATG-3', rev: 5'-AACACCCTTTCCAAATCCTCA-3'; hEMC-7 qPCR: fw: 5'-AAAGGAGGTAGTCAGGCCG-3', rev: 5'-GTTGCTTCACACGGTTTTCCA-3'; hRab7a qPCR: fw: 5'-TGACTGCCCCCAACACATTC-3', rev: 5'-TCCGTGCAATCGTCTGGAAC-3'; hRab7b qPCR: fw: 5'-CCTCCCTCCTTCACCAATA-3', rev: 5'-CAGTGTGGTCTGGTATTCCTCA-3'; hTPCN1 qPCR::5'-TCCCAAAGCGCTGAGATTAC-3', rev: 5'-TCTGGTTTGAGCTCCCTTTC-3'; hTPCN2 qPCR: fw: 5'-GTACCCCTCTTGTGTG GACG-3', rev: 5'-GGCCCTGACAGTGACAACTT-3'; mHprt qPCR: fw: 5'-GCTCGAGAT GTCATGAAGGAGAT-3', rev: 5'-AAAGAACTTATAGCCCCCTTGA-3'; mRab7 qPCR: fw: 5'-AGCCACAATAGGAGCGGACT-3', rev: 5'-CAAGTCTGTCGTCCACCATC-3'; mTpc2 qPCR: fw: 5'-TAAAGTACCGCTCCATCTACCA-3', rev: 5'-GCAGACGTTCGAGTAATACCAG-3'. Relative expression of target gene levels was determined by normalization against the house-keeping genes for the appropriate species.

## Western blotting (WB) and co-immunoprecipitation experiments

Buffer preparation and western blot experiments were performed as follows. Cells were washed with 1x PBS (Thermo Fisher) and pellets were collected. Total cell lysates were obtained by solubilizing in the buffer containing TRIS HCl 10 mM pH 8.0 and 0.2% SDS (Carl Roth) supplemented with fresh protease and phosphatase inhibitor Cocktails (Roche)[19,60]. For co-immunoprecipitation: HEK293 cells were plated on a 100 mm dish, and co-transfected at ca. 70% confluence with the respective plasmids via TurboFect (Thermo Fisher). 48 h post-transfection, cells were washed with ice-cold 1x PBS, and lysed with ice-cold lysis buffer containing: 1x PBS, 1 % Triton X-100, supplemented with 1x protease inhibitor (Merck) and 1x phosphatase inhibitor (Roche). The cell lysate was incubated for 30 min on ice, and vortexed. To clear the cell suspension, the lysate was centrifuged for 20 min at 17,000 g, 4 °C. Protein concentration of the supernatant was quantified via a Bradford assay (BioRad). Myc-tagged TPC2 and Myc-tag only overexpressed from the "empty vector" were immunoprecipitated with anti-Myc magnetic agarose beads (Proteintech). Beads were washed with lysis buffer and incubated with 1 mg of protein, rotating overnight at 4 °C. To ensure the removal of unspecific interaction partners, beads were washed three times with lysis buffer. After washing, the interaction partners were eluted from the beads by adding 2 x NuPAGE™ LDS - sample buffer (+ 10 % 2-mercaptoethanol) and incubating for 10 min at 95 °C. The "input sample" (30 μg protein sample) was treated the same way for preparation of SDS-PAGE and western blot. For analysis, the proteins were separated on a 7 % SDS-PAGE gel, transferred to a PVDF membrane (0.45 μm, Millipore). Membranes were developed via incubation with Immobilon Crescendo Western HRP substrate (Merck) and the Odyssey FC Imaging System (LI-COR) running the ImageStudio software v1.0.19. Proteins were quantified using unsaturated images for the ImageJ 1.52a software. The number of repeated blotting experiments corresponds to the number of biological replicates (n) included in the quantification of each western blot. The following primary antibodies were used at 1:1,000 concentration in 5% Bovine Serum Albumin (BSA) diluted in tris buffered saline supplemented with 0.5% Tween-20 (TBS-T). The following antibodies were used: CK1 (Cell Signaling Technology, 1:1000, cat. #2655), Phospho-Akt (Ser473) (Cell Signaling Technology, 1:1000, cat. #4058), Akt (Cell Signaling Technology, 1:1000, cat. #9272), MITF (Cell Signaling Technology, 1:1000, cat. #97800), GSK-3β (Cell Signaling Technology, 1:1000, cat. #9832), CREB and pCREB (Cell Signaling Technology, 1:1000, cat. #9197S and #9198S), Vinculin (Cell Signaling Technology, 1:1000, cat. #4650), GAPDH (Cell Signaling Technology, 1:1000, cat. #5174S), Anti-Mouse (Cell Signaling Technology, 1:10,000, cat. #7076), and Anti-Rabbit (Cell Signaling Technology, 1:10,000, cat. #7074), GAPDH (Cell Signaling Technology, 1:1000, cat. #5174), GSK-3β (Cell Signaling Technology, 1:1000, cat. #9832), MITF (Cell Signaling Technology, 1:1000, cat. #97800), Rab7 (Cell Signaling Technology, 1:1000, cat. #2094S), Rab7 (Cell Signaling Technology, 1:1000, cat. #9367S), β-Catenin (Cell Signaling Technology, 1:1000, cat. #9562), β-Actin (Santa Cruz Biotechnology, 1:1000, cat. #Sc-47778), anti-c-myc (Santa Cruz Biotechnology, 1:1000, cat. #sc-40), and anti-RFP (Proteintech, 1:1000, cat. #6g6), Non-phospho β-Catenin (Cell Signaling Technology, 1:1000, cat. #4270), Phospho-β-Catenin (Cell Signaling Technology, 1:1000, cat. #9561) (Supplementary Data 1).

## Cell proliferation and colony formation assays

Cells were seeded overnight in triplicates in flat-bottom 96-well microtiter plates (Sarstedt), using cells measured at 0-hrs as control blank. Proliferation was assessed using the CellTiter-Blue (Ctb) assay (Promega). Fluorescence was measured after 3 h of adding Ctb using the FLUOstar Omega running Reader at 560Ex/600Em (BMG LAB-TECH). For the colony formation assay, cells were seeded at a low-density of 2000 cells per well in 6-well plates and grown over 3 weeks in the incubator at 37 °C with 5% CO$_2$. Then, cells were fixed and stained with 0.5% crystal violet (Merck) in methanol (Carl Roth). Images were analyzed using ImageJ 1.52a software and the ImageJ plugin downloaded from and described in Guzmán et al.[61].

## Invasion and migration

Melanoma cells were seeded on 24-well transwell permeable supports polycarbonate membranes (Corning, 3421). The cells were pre-silenced or pre-stimulated with compounds then were additionally directly stimulated. The upper chambers contained the cells in serum-free medium and +/- compound. While the lower wells contained the chemotactic agent (10% FBS) and +/- compound. The distinction between the migration and invasion assay is the pre-coating the transwells for the latter with matrigel basement membrane matrix (Corning).

## LysoTracker Red staining

For confocal microscopy analysis cells were grown overnight in collagen-coated 8-well μ-slides (ibidi, Graefelfing, Germany). Subsequently, cells were left for 1 h at RT incubating with PBS containing 200 nM LysoTracker Red. Then cells were fixed in 4% PFA and nuclei were counterstained with Hoechst 33342 (100 μg/ml). Samples were mounted in FluorSave Mounting medium, sealed with glass coverslips, and analyzed on a Leica TCS SP8 confocal microscope (Leica, Wetzlar, Germany) and Las X software (Leica).

## FITC dextran accumulation

Cells were grown in 12-well plates overnight and incubated with 200 μg/ml FITC-dextran (20 kDa) (Sigma-Aldrich) for 90 min. After trypsinization, cells were collected by centrifugation, washed, and resuspended in PBS. Flow cytometry experiments were performed on a ThermoFisher Attune NxT. Fluorescence intensity of FITC-dextran was analyzed using the FITC channel. Data were evaluated using FlowJo 10.0 (https://www.flowjo.com/solutions/flowjo/downloads).

For FITC-Dextran accumulation evaluation via confocal microscopy, cells were grown overnight in collagen-coated 8-well μ-slides (ibidi) overnight, incubated with 200 μg/ml FITC-dextran (25 kDa) (Sigma-Aldrich, #FD20S) (2 h). After fixation (4% PFA, 10 min, RT), samples were mounted with FluorSave mounting medium (Merck Millipore), covered with glass cover slips and imaged.

## FITC dextran release

Cells were seeded into 96-well plates and stimulated with 200 μg/ml FITC-dextran (20 kDa) (Sigma Aldrich). After 24 h, cells were washed and incubated with 50 mM CaCl$_2$ in Phenol Red free DMEM (Pan Biotech) as indicated. After diluting 1:10 in PBS, fluorescence intensity was measured in the supernatants using the Infinite® 200 Pro Tecan Plate reader (Tecan Trading AG) (485/535 Ex/Em).

## In vivo experiments

B16F10luc WT, TPC2 KO or Rab7a KO cells were injected into 5-6 week old, female C57Bl/6-Tyr mice (Envigo). For the ectopic tumor growth experiment 2 x 10$^6$ B16F10-luc WT, TPC2 KO or Rab7a KO cells were subcutaneously injected into the flank. For the tumor dissemination experiment 2 x 10$^5$ of the same lines were injected into the tail vein. Treatment of mice in the ectopic model with vehicle control, Rab7 agonist ML-098 (0.04 mg/g) or TPC2-A1-P (0.02 mg/g) was performed daily, starting right after implantation of cells. Compounds were dissolved in a solution containing 85% PBS, 10% Solutol-15® and 5% DMSO and injected intraperitoneally with a volume of 100 μL. Bioluminescence imaging (IVIS™) was performed regularly after implantation of cells following intraperitoneal injection of 6 mg/mL luciferin per mouse. Prior to imaging, mice were put under anesthesia with 2.5% isoflurane in oxygen. Imaging of mice was performed in ventrodorsal position and mice were kept under narcosis with 1.5% isoflurane in

oxygen. Hypothermia was prevented by a heating plate (37 °C). The tumor signal per defined region of interest was calculated as photons/second/cm (total flux/area) using the Living Image 4.4 software (Perkin Elmer). Abortion criteria was set to a maximal tumor burden of $10^{11}$ total flux/area, which equals a tumor size of $1200\,mm^3$ with a diameter of 14,5 mm. All research performed complies with all relevant ethical regulations. Animals were used under animal protocols approved by the government (Regierung von Oberbayern, ROB-55.2-2532.Vet_02-22-5), and University of Munich (LMU) Institutional Animal Care Guidelines. Mice were housed in rooms maintained at constant temperature (20-24 °C) and humidity (45-65%) with a 12-hour light cycle. Animals were allowed food and water ad libitum.

### Immunohistochemistry

For immunohistochemistry experiments, tumors were excised from the mice and immediately fixed in 10% neutral-buffered formalin for 48 h. Subsequently, the tissue samples were transferred to 95% ethanol for 48 h, followed by 70% ethanol for an additional 48 h. The tumors were then embedded in paraffin using an embedding center (TN1700, Tanner Scientific). Paraffin-embedded tumors were sectioned into 10 μm thick slices using a microtome (HM355, Thermo Scientific). Duplicate sections were mounted onto glass slides. For stainings tissue sections were deparaffinized and rehydrated through a series of xylene and ethanol washes. Antigen retrieval was performed by incubating the sections in citrate buffer (pH 6.0) at 95 °C for 20 min. Endogenous peroxidase activity was quenched by treating the sections with 7.5% hydrogen peroxide for 10 min. Primary antibodies were applied overnight at 4 °C: anti-MITF (Cell Signaling Technology, #12590, diluted 1:150) and anti-β-Catenin (Cell Signaling Technology, #8480, diluted 1:150). The following day, sections were washed with phosphate-buffered saline (PBS) and incubated with a secondary antibody (Alexa Fluor 488-conjugated, Invitrogen, A-11008, diluted 1:400) and Hoechst 33342 (diluted 1:400) for one hour at room temperature. After final washing steps, tissue sections were mounted using Fluorsave mounting medium (Merck, #345789) and coverslipped. Imaging was performed using a Leica DMi1 inverted microscope equipped with a MC120HD camera (Leica, Wetzlar, Germany). Quantification of positively stained cells was conducted using QuPath software (version 0.4.4).

### Human tissue samples

We obtained human tissue samples form 10 metastatic lymph nodes from melanoma patients and healthy lymph nodes that had already been archived previously at the Department of Pathology of the LMU Munich. The formalin-fixed, paraffin embedded, samples were selected by the Department of Pathology and contained equal samples from male and female donors. The experiments are in accordance with ethical standards of the responsible committee on human experimentation (written approval from the ethics committee of the LMU Hospital, Munich, number 23-0119; informed consent was obtained from all participants) and with the Helsinki Declaration of 1975, as revised in 2000. Tissue samples were pretreated with Target Unmasking Fluid (Pan Path, Z000R0000) (Rab7) or Pro Taqs IV Antigen-Enhancer (Quartett, 401602392) (MITF) and incubated with primary antibody 1:160 (Rab7, Cell Signaling, #95746) or 1:100 (MITF, Agilent, M3621) for 60 min at RT. Subsequently ImmPRESS Anti-Mouse IgG Polymer Kit (Vector, MP-7402) was used for detection. The chromogen of choice is AEC (Bio SB, BSB 0011) and tissues were counterstained with Hematoxylin Gill's Formula (Vector, H-3401).

### Statistical analysis

Analysis of the recordings was completed with Origin9, whilst statistics were generated with GraphPad Prism from a minimum of three repetitive, independent experiments. In addition, all the error bars depicted are ± SEM. Statistical comparison and significance were made using one-way ANOVA, two-way ANOVA followed by Bonferroni multiple comparisons test, or Student's Unpaired t-test depending on the statistical analysis required in each Fig., described in further detail in the corresponding Fig. legends.

### Reporting summary

Further information on research design is available in the Nature Portfolio Reporting Summary linked to this article.

## Data availability

All data supporting the findings from this study are available within the manuscript and its Supplementary Information. Source data are provided with this paper or with this submission (Figshare DOI (https://doi.org/10.6084/m9.figshare.25907827)).

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

## Acknowledgements

This work was supported, in part, by funding of the German Research Foundation (SFB/TRR152 P04 to C.G., P06 to C.W.-S., P12 to M.B., and P15 to T.G., SFB1328 A21 to C.G., DFG GR4315/2-2 to C.G. and BA238/3-2 to K.B., DFG GR4315/4-1 to C.G., and INST 192/543-1 FUGG to C.W.-S.). C.-C.C. was supported by the Ministry of Science and Technology

(MOST 110-2320-B-002-022), National Taiwan University (NTU-111L7826) and National Health Research Institutes (NHRI-EX111-11119SC). We thank Jesslyn Yonathan for technical support and Prof. Dr. Thorsten Kessler for providing the pDEST40-2XHA-wtSrc plasmid. Figures 7m and 8 were created in Biorender.

## Author contributions

C.A., R.T., R.D., L.O., E.-M.W., V.K., C.F., Y.S., A.S.R. and C.-C.C. designed experiments and collected and analyzed data. R.D. and R.T. performed endolysosomal patch-clamp experiments. R.D. performed co-immunoprecipitation experiments. L.O., J.B., K.B., T.F. and J.R. carried out animal experiments. C.F. performed FRET experiments. V.K. provided $Ca^{2+}$ imaging experiments. M.S. and G.C. provided melanoma cell lines. T.F., T.G., M.B., C.W.-S. and K.B. provided funding and commented on the manuscript. S.K. and D.M. provided and analyzed human melanoma and control samples. C.G. provided funding, coordinated research, designed the study, analyzed data, designed figures, and wrote the manuscript. All of the authors discussed the results and commented on the manuscript.

## Funding

## Competing interests

The authors declare no competing interests.
