## [Transparent Peer Review file · Nature Communications]

Rab7a is an enhancer of TPC2 activity regulating melanoma progression through modulation of the GSK3 β / β -Catenin/ MITF-axisEditorial Note: Parts of this Peer Review File have been redacted as indicated to remove third-party material where no permission to publish could be obtained.

REVIEWER COMMENTS

Reviewer #1 (Remarks to the Author):

This is an interesting study looking at the role of TPC2 and Rab7a as regulators of melanoma tumorigenesis and metastasis parameters. Previously studies have identified a role for TPC2 in melanoma and also shown that Rab7a interacts with TPC2. This study goes beyond such previous studies in demonstrating a functional link between TPC2 and Rab7a and also that TPC2 modulates the Wnt signaling pathway, in particular GSK3 β -mediated degradation of MITF. The study also has an in vivo angle to complement the in vitro findings.

A few specific points.

1. Figure 6, a, g shows MITF at 62kDa whereas in c, e it is 70kDa. Can the authors explain the discrepancy?
2. In Figure 7, the TPC2 KO tumor seem to be bigger at day 7 as judged by its red luminescence . There are no images for the 2nd group in the line graph (WT + TPC2-A1-P). The experiment was performed only for 7 days. It would good if it was allowed to run for longer.
3. The study did not show any evidence of in vivo migration in invasion only growth for 7 days. The rest is in vitro. Can the authors discuss whether migration was found in vivo?

Reviewer #2 (Remarks to the Author):

Key results

The authors present data suggesting that Rab7a acts as an effector of TPC2 activity in MITF high melanoma cells. TPC2, a two pore cation channel on melanosomes/endolysosomes, directly interacts via protein-protein interactions with Rab7a, a small GTPase, which when knocked down or out decreases TPC2 activity resulting in a decrease in MITF and beta-catenin. TPC2 has been previously reported to interact with Rab7a. Here the authors present data suggesting the interaction between TPC2 and Rab7a regulates the expression of MITF, a master transcription regulator in melanoma, by regulating the degradation of GSK3beta. Knocking down either TPC2 or Rab7a was found to decrease MITF expression, causing a decrease in proliferation and invasion in melanoma cells with high MITF. Melanoma cells with low MITF did not respond to alteration in TPC2 and Rab7a by altering invasion of proliferation. The authors have done a good job of explaining their results and impact in context of what is currently known in the field.

Validity and robustness of data interpretations and conclusions: The authors have used multiple assays, cell lines, and methods to evaluate the activity, and phenotypic impact of the interaction of TPC2 and Rab7a.

Significance: The authors have clearly explained the impact of their findings in the discussion. These finding reveal the role of Rab7a, a small GTPase, in regulating the expression of MITF which plays an important role in melanoma progression.

Data and Methodology: Data is clearly analyzed and interpreted to generate appropriate conclusions. A few minor comments below regarding quantification of data under suggested

improvements.

Analytical approach: There is enough detail in the methods for the work to be reproduced.

Suggested improvements:

Figure 1 b. Why are the western bands shown in duplicated for each cell line? Does there need to be a label for the duplicate bands for each cell line?

Figure S2g – needs a y-axis label

Figure S3. Genotyping gels for TPC2 KO – only the AF22 clone is shown, and only the C1x17 clones is shown for Rab7a clone in SK-MEL-5 cells.

Figure 4a needs a y-axis label.

Figure 5a-d aren't labeled.

Figure 5. Authors mention that the data for TPC2 and Tab7 KD are conserved in their effects on proliferation and invasion however with A375 the KDs had a significant effect on proliferation but not on invasion.

Figure 6C. which cell lines correspond to the bands shown for the WT and Rab7a KO?

The authors suggest inhibiting Rab7a or TPC2 decreases GSKB expression, however, in the Rab7aKO (Figure 6C) there does not appear to be a significant difference in protein expression of GSKbeta compared to WT.

Figure 6D and F. What samples or bands are these bar graphs quantitating? There are five WT bands and 5 Rab7a KO bands for westerns in 6c, but there are more than five data points for each group in the bar graph in Figure 6d.

Figure 7O. It is difficult to tell the difference between the WT and Rab7a KO+TPC2-A1-P color and symbol on the graph. Maybe alter the symbol shape for one or make one a dashed line.

Reviewer #3 (Remarks to the Author):

The authors revisited the TPC2-RAB7A interaction and studied its role in melanoma tumorigenesis. Although the electrophysiology data are impressive, there are several limitations in experimental designs and results that fully support the authors' claim.

Major Comments:

1. WNT Signaling: The manuscript lacks WNT signaling-related data, necessitating major edits to the title and text or the inclusion of additional experimental results to justify their relevance.

2. Molecular Mechanism: The mechanism of the TPC2-RAB7A axis remains unclear. Clarification is needed on how binding/protein interaction is related to activity regulation.

3. Correlation vs. Causality: The authors frequently attribute correlation to causality, particularly regarding MITF and WNT signaling in relation to TPC2 and RAB7A. Establishing causation requires additional rescue or GOF/LOF data.

4. Redundancy: The authors' previous publications have already reported TPC2's roles in cancer cell migration, invasion, and proliferation, diminishing the novelty of the current manuscript.

5. In Vivo Data: Improvement is needed in the presentation of in vivo data, especially regarding Fig. 7n and o, such as in IHC.

Minor Comments:

1. Direct Effect of Rab7a on TPC2: This study did not demonstrate a direct interaction between Rab7a and TPC2.

2. Molecular Mechanism: Using a small molecule inhibitor of Rab7 is insufficient to elucidate the molecular mechanism.
3. mRNA vs. Protein: Clear differentiation between mRNA and protein is necessary in describing the text.
4. Previous Reports: The Rab7-TPC2 interaction was previously reported in PNAS 2016.
5. Endogenous Interaction: Was the endogenous interaction between Rab7a and TPC2 demonstrated?
6. Experimental Design: Justification is needed for not using melanocytes or melanoma in electrophysiology experiments.
7. Fig. 6. The MITF story is another biased stretch. Again, correlation does not guarantee causation. In other words, picking MITF over many other proteins or molecules involved in melanoma tumorigenesis and WNT signaling was not justified or convincing. Descriptive results are insufficient to demonstrate that TPC2-RAB7A's impact is mediated by MITF without rescue results (using MITF WT as well as mutants [if available]).
8. WNT Signaling: WNT signaling is frequently hyperactivated in melanoma. Adding the following to the Discussion is highly recommended (only if the authors want to include the WNT signaling in this manuscript): Why is an additional layer of regulation (positive) of the WNT signaling necessary for melanoma tumorigenesis? In other words, is it also possible that the impact of the TPC2-RAB7A axis on melanoma tumorigenesis is WNT signaling-independent? What is the rationale for the authors to consider only WNT signaling to explain the TPC2-RAB7A axis in melanoma tumorigenesis?

Reviewer #4 (Remarks to the Author):

An interesting study by Abrahamian et al. about the role of small GTPase RAB7A as an oncogene in melanoma cancer, in vitro and in vivo. The work was well written, and I appreciated how the experiments were designed and how each finding fueled the next question.

I recommend this paper be published, however there are several major concerns in this work as shown in following comments:

- The authors declared that RAB7A is an effector of TPC2, but they did not demonstrate the effect on endosomal function. I suggest to perform in vitro and ex vivo experiments to observe the effect of TPC2 KO/KD and RAB7A KO/KD on the lysosomal pathway, acidification, and function (immunofluorescence for lysosomal activity, size, and number; valuation of expression levels of endosomal trafficking proteins; analysis with TEM microscopy of endosomal compartments). Indeed, to the best of my knowledge, the KO of RAB7A is lethal in vivo. Thus, it is plausible that in the cell lines, RAB7 KO induces compensatory effects and other pathway activation.
- Alfonso-Curbelo declared that, in the same cell lines used in this work, regulation of RAB7A is MITF-independent demonstrating that levels of RAB7 are different in the progression of melanoma. In particular, he demonstrated that the oncogenic transcription factor Myc is activated and induces a strong overexpression of Rab7. How do the authors justify MITF dependence in the same cell lines? I think that the authors should demonstrate that silencing MITF induces RAB7 and TPC2 downregulation and viceversa. Moreover, to validate their hypothesis they should demonstrate the increase of GSK3B degradation when RAB7A and TPC2 are upregulated and viceversa. Indeed, the authors "postulated" this mechanism, but they do not demonstrate it.

Minor: The authors compared the expression of RAB7A to the expression of RAB7B. I think that this comparison is useless because these two proteins are not isoforms, their identity is limited to 50% and they do not share the RabF and RabFS motifs. Indeed, these proteins have also very different functions, have differential localization and control different steps of transport. RAB7B is involved in the control of endosomes to Golgi transport, being localized both to the Trans Golgi Network and late endosomes.

REVIEWER COMMENTS

Reviewer #1 (Remarks to the Author):

This is an interesting study looking at the role of TPC2 and Rab7a as regulators of melanoma tumorigenesis and metastasis parameters. Previously studies have identified a role for TPC2 in melanoma and also shown that Rab7a interacts with TPC2. **This study goes beyond such previous studies in demonstrating a functional link between TPC2 and Rab7a** and also that TPC2 modulates the Wnt signaling pathway, in particular GSK3 β -mediated degradation of MITF. The study also has an in vivo angle to complement the in vitro findings.

Authors: We thank the reviewer for deeming our results interesting and for their recognition that our study extends beyond the scope of previous research.

A few specific points.

1. Figure 6, a, g shows MITF at 62kDa whereas in c, e it is 70kDa. Can the authors explain the discrepancy?

Authors: We apologize for the mistake, there has been indeed a mislabelling. The molecular weight of the MITF cell signalling technology antibody is 62 kDa. We have now ensured uniformity across all figures.

2. In Figure 7, the TPC2 KO tumor seem to be bigger at day 7 as judged by its red luminescence. There are no images for the 2nd group in the line graph (WT + TPC2-A1-P). The experiment was performed only for 7 days. It would good if it was allowed to run for longer.

Authors: We agree with the reviewer. We have now performed more in vivo experiments to strengthen our claims and also analysed endpoint results (day 14). In particular, we show now that TPC2 KO and Rab7 KO in vivo result in significant reduction of tumour weight compared to WT, in line with our in vitro findings on tumour hallmarks after either TPC2 or Rab7 KO/KD. In addition, the reduced tumour weight after Rab7 KO cell injection could be partially reversed with TPC2-A1-P agonist treatment, reaching values that were not significantly different from WT at 14 days. This is in line with the bioluminescence data showing significantly increased bioluminescence signal after TPC2-A1-P treatment in Rab7 KO versus vehicle treated Rab7 KO control already at day 7 (Fig.7n). By contrast, application of a Rab7 agonist, ML-098 did have no effect on tumour weight in TPC2 KO, further supporting the claim from our in vitro (rescue) experiments that TPC2 OE or activation can reverse the effect in Rab7 KO while Rab7 OE or activation cannot rescue TPC2 KO. Specifically, we now show the isolated tumours at day 14 when the mice were sacrificed (new Fig.7m-q).

3. The study did not show any evidence of in vivo migration in invasion only growth for 7 days. The rest is in vitro. Can the authors discuss whether migration was found in vivo?

Authors: We now provide additional data on migration behavior in vivo as the reviewer requested. We performed a tumour dissemination assay in which tumour cells are allowed to migrate to distant organs after tail vein injection. In accordance with our reported in vitro data (see also new Suppl. Fig.S7h-I and below) and in line with previous publications on the impact of TPC2 KO on cancer cell migration (Müller et al., Cell Chem Biol., 2021, Nguyen et al., Cancer Res., 2017), dissemination of cancer cells is impaired upon knockout of TPC2 and also Rab7. Unfortunately, we have no approval for testing the compounds, TPC2-A1-P and Rab7 agonist ML-098 in this model. A de novo application with the Bavarian Government would take >12 months.

Reviewer #2 (Remarks to the Author):

Key results

The authors present data suggesting that Rab7a acts as an effector of TPC2 activity in MITF high melanoma cells. TPC2, a two pore cation channel on melanosomes/endolysosomes, directly interacts via protein-protein interactions with Rab7a, a small GTPase, which when knocked down or out decreases TPC2 activity resulting in a decrease in MITF and beta-catenin. TPC2 has been previously reported to interact with Rab7a. Here the authors present data suggesting the interaction between TPC2 and Rab7a regulates the expression of MITF, a master transcription regulator in melanoma, by regulating the degradation of GSK3beta. Knocking down either TPC2 or Rab7a was found to decrease MITF expression, causing a decrease in proliferation and invasion in melanoma cells with high MITF. Melanoma cells with low MITF did not respond to alteration in TPC2 and Rab7a by altering invasion or proliferation. **The authors have done a good job of explaining their results and impact in context of what is currently known in the field.**

Validity and robustness of data interpretations and conclusions: The **authors have used multiple assays, cell lines, and methods** to evaluate the activity, and phenotypic impact of the interaction of TPC2 and Rab7a.

Significance: The authors **have clearly explained the impact of their findings** in the discussion. These findings reveal the role of Rab7a, a small GTPase, in regulating the expression of MITF which plays an important role in melanoma progression.

Data and Methodology: **Data is clearly analyzed and interpreted** to generate appropriate conclusions. A few minor comments below regarding quantification of data under suggested improvements.

Analytical approach: There is **enough detail in the methods** for the work to be reproduced.

Authors: We appreciate the reviewer's positive and supportive comments regarding validity and robustness of our data and data interpretations and conclusions as well as significance, methodology and the analytical approach of our study.

Suggested improvements:

Figure 1 b. Why are the western bands shown in duplicated for each cell line? Does there need to be a label for the duplicate bands for each cell line?

Authors: We appreciate the reviewer's question regarding Figure 1b and the presentation of Western blot bands for each cell line. The decision to display duplicate bands for each cell line was made to highlight the consistency and reproducibility of our results across independent biological replicates. We would also like to draw the reviewer's attention to Figure 6a, where a different western blot is shown, depicting only one protein sample from each cell line. We could replace Figure 1b with Figure 6a for ease or we keep both as the reviewer prefers.

Figure S2g – needs a y-axis label

Authors: We apologize for the mistake. We have added a labelling now "Current Density (pA/pF)".

Figure S3. Genotyping gels for TPC2 KO – only the AF22 clone is shown, and only the C1x17 clones is shown for Rab7a clone in SK-MEL-5 cells.

Authors: We would like to clarify that we generated one clone each of Rab7a KO (C1x17) and TPC2 KO (AF22) in this study. The genotyping results for the two other clones of TPC2 KO, AF9 and AF19 have been published previously in Yuan et al., 2022, Nature Communications, Supplementary Figure 2b. We have added a comment in the Results section referring to and citing this paper.

Figure 4a needs a y-axis label.

Authors: We apologize for the mistake, we have added the labels accordingly.

Figure 5a-d aren't labeled.

Authors: We apologize for the mistake, we have added the labels accordingly.

Figure 5. Authors mention that the data for TPC2 and Rab7 KD are conserved in their effects on proliferation and invasion however with A375 the KDs had a significant effect on proliferation but not on invasion.

Authors: Yes. This is correct. TPC2KD in A375 only showed an effect on proliferation but not on invasion. The same was true for Rab7KD. Rab7KD of A375 also showed only an effect on proliferation but not on invasion. Thus, also in this line both Rab7 and TPC2KD results are concordant. Nevertheless, this result is surprising as based on the MITF expression a proliferation result like in SK-MEL103 or SK-MEL147 would have been expected i.e., no effect. Nevertheless, we did not simply remove this line only because it did not fit to the rest of the data. We acknowledge that the different lines we used differ in certain known and possibly unknown aspects e.g., the A375 cell line is AXL-predominant and harbors the BRAFV600E mutation, similar to SK-MEL-5, SK-MEL-19, and UACC-62. By contrast the melanoma lines SK-MEL-147 and SK-MEL-103 are BRAF WT and harbor the NRASQ61R mutation. This diversity in mutational profiles across cell lines can contribute to variations in cellular responses. We speculate that additional differences in the mutational background of A375 may influence the observed outcome. However, please note that in general all the tested lines very well correlate in phenotypes after either TPC2 KO/KD or Rab7 KO/KD.

Figure 6C. which cell lines correspond to the bands shown for the WT and Rab7a KO?

Authors: We appreciate the reviewer's observation, and we have addressed this by adding the information to the Fig. legend. The cell line utilized was SK-MEL-5, which has high expression levels of Rab7a, TPC2, and MITF. Please note that in new Fig.6 the sequence of data presented

has changed (e.g. previous Fig.6c and d are now f and g; g shows the statistics corresponding to the representative blots shown in f). See also next Point.

The authors suggest inhibiting Rab7a or TPC2 decreases GSK3 β expression, however, in the Rab7aKO (Figure 6C) there does not appear to be a significant difference in protein expression of GSKbeta compared to WT.

Authors: Our data demonstrate that the KO of Rab7a or TPC2 results in increased expression of GSK3 β protein while decreasing β -catenin expression compared to their WT counterparts (see statistics in new Fig.6g belonging to representative blots in f as well as Fig.6i, which displays the statistics belonging to representative blots in h).

The absence of Rab7 and TPC2 disrupts the ESCRT machinery, preventing the sequestration of the destruction complex (i.e., GSK3 β) into late endosomes and leading to elevated GSK3 β expression in melanoma. Consequently, β -catenin undergoes increased phosphorylation (by GSK3 β), rendering it unstable and unable to translocate to the nucleus. As a result, we observe a subsequent reduction in MITF levels. This is in line with recent data published by Ploper et al., PNAS, 2015. See also next two Points.

In addition, we have now performed further rescue experiments, looking directly at the effect of Rab7 on GSK3 β in Rab7 and TPC2KOs. Increased GSK3 β levels in Rab7 KO could be reduced again when overexpressing either Rab7aWT or Rab7aQ67L (Fig. S6a). Vice versa however, increased GSK3 β levels in TPC2 KO could not be reversed by overexpressing either Rab7aWT or Rab7aQ67L (Fig. S6b). These findings corroborate the hypothesis that Rab7a acts as an effector of TPC2 but not vice versa and that both Rab7 and TPC2 KO show a strong regulatory effect on GSK3 β .

We could also demonstrate that OE of GSK3 β further reduces proliferation of Rab7a KO as well as TPC2 KO SK-MEL-5 cells (new Fig. S6c).

Figure 6D and F. What samples or bands are these bar graphs quantitating? There are five WT bands and 5 Rab7a KO bands for westerns in 6c, but there are more than five data points for each group in the bar graph in Figure 6d.

Authors: We thank the reviewer for their comment (see also Points above). Old Fig.: Figure 6c displays the representative western blot for each dataset. While the bar graphs in Figure 6d quantify the expression levels observed in multiple biological replicates, which accounts for the presence of more than five data points in each group. These repeated additional data points provide a more comprehensive representation of the experimental results, capturing variability across multiple independent experiments. Please note that in new Fig.6 the sequence of data presented has changed (i.e. previous Fig.6c and d are now f and g). This is because we have now incorporated human healthy and melanoma tissue data analysed for MITF expression shown in d and e.

Figure 7O. It is difficult to tell the difference between the WT and Rab7a KO+TPC2-A1-P color and symbol on the graph. Maybe alter the symbol shape for one or make one a dashed line.

Authors: We agree with the reviewer's comment. We have now redesigned this part of Fig.7 as we have performed more in vivo experiments in the meantime and importantly now also analysed endpoint results (day 14). In particular, we show now that TPC2 KO and Rab7 KO in vivo result in significant reduction of tumour weight compared to WT, in line with our in vitro findings on tumour hallmarks after either TPC2 or Rab7 KO/KD. In addition, the reduced tumour weight after Rab7 KO cell injection could be partially reversed with TPC2-A1-P agonist treatment, reaching values that were not significantly different from WT at 14 days. This is in line with the bioluminescence data showing significantly increased bioluminescence signal after TPC2-A1-P treatment in Rab7 KO versus vehicle treated Rab7 KO control already at day 7 (Fig.7n). By contrast, application of a Rab7 agonist, ML-098 did have no effect on tumour weight in TPC2 KO, further supporting the claim from our in vitro (rescue) experiments that TPC2 OE or activation can reverse the effect in Rab7 KO while Rab7 OE or activation cannot rescue TPC2 KO. Specifically, we now show the isolated tumours at day 14 when the mice were sacrificed (new Fig.7m-q).

We now also provide additional data on migration behavior in vivo. We performed a tumour dissemination assay in which tumour cells are allowed to migrate to distant organs after tail vein injection. In accordance with our reported in vitro data (wound healing assay, see also new Suppl. Fig.S7h-I and below) and in line with previous publications on the impact of TPC2 KO in cancer cell migration (Müller et al., Cell Chem Biol., 2021, Nguyen et al., Cancer Res., 2017), dissemination of cancer cells is impaired upon knock-out of TPC2 or Rab7. Unfortunately, we have no approval for testing the compounds, TPC2-A1-P and Rab7 agonist ML-098 in this model. A de novo application with the Bavarian Government would take >12 months.

Wound healing assay

Reviewer #3 (Remarks to the Author):

The authors revisited the TPC2-RAB7A interaction and studied its role in melanoma tumorigenesis. Although the **electrophysiology data are impressive**, there are several limitations in experimental designs and results that fully support the authors' claim.

Authors: We appreciate the reviewer's opinion that our "electrophysiology data are impressive", clearly a major and novel point of our study as we demonstrate for the first time that Rab7a is a molecular enhancer of the TPC2 channel activity using lysosomal patch-clamp electrophysiology.

We have taken the reviewer's comments on the other experiments very seriously and provide detailed responses and new experimental data below. We would explicitly like to thank the Reviewer for the very important and very valid comments made to improve our MS.

Major Comments:

1. WNT Signaling: The manuscript lacks WNT signaling-related data, necessitating major edits to the title and text or the inclusion of additional experimental results to justify their relevance.

Authors: We thank the reviewer for their comment. First, we would like to clarify that we only want to claim that Wnt signaling pathway components such as GSK3 β and β -Catenin are affected by Rab7a/TPC2 KO/KD, not necessarily the entire signaling cascade and/or even receptor signaling. We have examined two such key components of the Wnt signaling pathway, GSK3 β and β -Catenin ([https://www.jidonline.org/article/S0022-202X\(15\)33674-5/fulltext](https://www.jidonline.org/article/S0022-202X(15)33674-5/fulltext)). We have now performed additional rescue experiments, see also Point 3, looking directly at the effect of Rab7 on GSK3 β in Rab7 and TPC2KO. Increased GSK3 β levels in Rab7 KO could be reduced again when overexpressing either Rab7aWT or Rab7aQ67L (Fig. S6a). Vice versa however, increased GSK3 β levels in TPC2 KO could not be reversed by overexpressing either Rab7aWT or Rab7aQ67L (Fig. S6b). These findings corroborate the hypothesis that Rab7a acts as an effector of TPC2 but not vice versa and that both Rab7 and TPC2 KO show a strong regulatory effect on GSK3 β and thus β -Catenin, two central Wnt-signaling components.

We could also demonstrate that OE of GSK3 β further reduces proliferation of Rab7a KO as well as TPC2 KO SK-MEL-5 cells (new Fig. S6c).

As mentioned, in addition to GSK3 β increase in Rab7 and TPC2KO, β -Catenin was significantly reduced in both KOs, further corroborating an important regulatory role of TPC2 and Rab7a on these Wnt pathway components. Important additional components of the β -Catenin destruction complex in the Wnt pathway are e.g., Axin, APC, or CK1 α . We have performed now additional WB experiments for one of these components, CK1 α (Casein kinase 1 α), which showed significantly increased expression like GSK3 β in Rab7KO and TPC2KO cells. See new Fig.S9 and below.

We are well aware that other pathways can impact GSK3 β and hence β -Catenin and MITF e.g., cKit/RAS/PI3K/Akt, with Akt directly inhibiting GSK3 β but also other pathways that affect MITF more directly. See more on this in Points 7 and 8 (Minor) below.

2. Molecular Mechanism: The mechanism of the TPC2-RAB7A axis remains unclear. Clarification is needed on how binding/protein interaction is related to activity regulation.

Authors: See also Points 1 and 2 below (Minor).

We propose a direct/physical and transient/reversible interaction between Rab7a and TPC2 based on the following points:

1) The direct interaction involving the N-terminus of TPC2 was already shown in Lin-Moshier et al., 2014 where a CoIP was performed using GFP-Rab7a and myc-tagged NH₂- or COOH-terminal domains of TPC2, revealing that GFP-Rab7a associates with Nter-TPC2 but not with Cter-TPC2. Furthermore, point mutations within the TPC2 NH₂-terminal domain, targeting a motif conforming to the consensus of a GTPase binding and trafficking module, significantly reduce Rab7 association with the NH₂ terminus of TPC2. Importantly, truncation of this region abolished Rab7 binding to TPC2 altogether.

2) We additionally performed two-hybrid FRET experiments for real-time and quantitative information about Rab7a and TPC2 protein interaction in living cells with high spatial and temporal resolution. Our data showed that the coexpression of TPC2 with the constitutively active mutant, Rab7aQ67L, resulted in 51% FRET efficiency while co-expression with the dominant negative Rab7a variant Rab7aT22N yielded only 10%, suggesting a relevant role of the GTP-logged form of Rab7a in modulating TPC2 activity, i.e. we elucidated that mechanistically the GTP-logged form of Rab7a participates in the effect on TPC2.

To further support this claim we performed new experiments with GTP to enhance the activity of endogenous Rab7 directly. See new GTP data below and in new Suppl. Fig. S2. Preincubation (5min) with GTP enhances TPC2 activity, supporting the notion that GTP increases the amount of endogenous GTP-logged Rab7a to modulate TPC2.

3) In line with the data above our GCaMP based Ca^{2+} imaging experiments have shown that TPC2 is enhanced when coexpressed with Rab7aWT, but that there is a further enhancement of TPC2 activation when coexpressed with Rab7aQ67L (GTP-logged). Our electrophysiology data show an equally enhanced activation of TPC2 with Rab7aWT and Rab7aQ67L, suggesting a potential saturation of the current; however and importantly in both Ca^{2+} -imaging and patch-clamp experiments loss of current activity in Rab7aT22N (GDP-logged) strongly argues for a relevant role of the GTP-logged, activated form of Rab7a in TPC2 activity regulation.

To further elucidate the interaction mechanism on the protein level, we would need Cryo-EM data. Cryo-EM experiments (Rab7+TPC2) with Prof. Dr. Christine Ziegler, University of Regensburg are currently being performed. Certainly, if successful this would add additional, precious information on the mechanism. However, we feel such a project is likely to take much more time and would be beyond the scope of this MS.

3. Correlation vs. Causality: The authors frequently attribute correlation to causality, particularly regarding MITF and WNT signaling in relation to TPC2 and RAB7A. Establishing causation requires additional rescue or GOF/LOF data.

Authors: We thank the reviewer for their comment. We have now performed additional rescue experiments in the CRISPR/Cas9 generated SK-MEL-5 Rab7-KO line as outlined above (Point 1). While the KOs of Rab7 and TPC2 have shown increased GSK3 β expression, overexpressing either Rab7aWT or Rab7aQ67L in Rab7 KO has conversely shown decrease of GSK3 β expression. Importantly and in line with our proliferation and invasion data, the overexpression of Rab7a in TPC2 KO has not shown any differences in GSK3 β protein levels, further confirming that Rab7a OE cannot rescue in absence of TPC2 (see new Suppl. Fig.6a,b). We could also demonstrate that OE of GSK3 β further reduces proliferation of Rab7a KO as well as TPC2 KO SK-MEL-5 cells (new Fig. S6c). For the respective figures see also Point 1.

Our data show that TPC2/Rab7 KO/KD results in reduced proliferation, migration, invasion and tumour growth, with most efficient effects seen in high MITF expressing melanoma lines. At the same time, we observed increased GSK3 β , reduced β -Catenin and reduced MITF levels in these lines after TPC2/Rab7 KO/KD or pharmacological block, suggesting that Rab7/TPC2 KO/KD/block is causative for this finding.

4. Redundancy: The authors' previous publications have already reported TPC2's roles in cancer cell migration, invasion, and proliferation, diminishing the novelty of the current manuscript.

Authors: We appreciate the reviewer's comment and acknowledge the existing body of literature in different cancers. It is correct that KD or KO of TPC2 has been shown previously to reduce cancer cell migration, invasion, and proliferation, e.g. in HCC. We have mentioned this in the MS. However, one important novel aspect of our study is that Rab7a is controlling TPC2 activity and that loss of Rab7a acts like TPC2 KD in cancer lines due to its strong regulatory effect on TPC2 activity, especially in those that express high Rab7a levels e.g., many melanoma lines,

Our data showing elevated expression levels of Rab7a and TPC2 in melanoma compared to other cancers (Fig.1) further emphasize their relevance specifically to melanoma pathogenesis. Furthermore, our findings indicate that these effects predominantly occur in melanoma cell lines with high MITF expression, establishing an axis where Rab7a enhances TPC2 activity, thereby promoting the proliferation, migration, and invasion of MITF+ melanoma. The identification of druggable Rab7a and TPC2 as upstream regulators of the melanoma oncogene MITF is significant given MITF's status as an undruggable oncogene. This discovery opens up new avenues for therapeutic intervention by targeting Rab7a and TPC2 in this subset of melanoma. Furthermore, high expression levels of Rab7a or TPC2 may serve as biomarkers indicating a potentially poor prognosis for patients or an increased risk to develop melanoma (TPC2 GOF carriers in combination with Rab7GOF may be at particular risk), thus paving the way for more tailored treatment and increased risk awareness.

To add to that, we have used the pharmacological inhibitor of TPC2, SG094, and have tested its effects in different melanoma lines. We have shown that inhibition of TPC2 in high MITF expressing lines such as SK-MEL-5, SK-MEL-19, or UACC-62 results in reduced proliferation, and in accordance reduced MITF protein levels after pharmacological TPC2 inhibition compared to control, confirming the KO data.

We would also like to point out that the three other reviewers acknowledged the novelty of our work explicitly and specifically appreciated the significance and impact of our study.

5. In Vivo Data: Improvement is needed in the presentation of in vivo data, especially regarding Fig. 7n and o, such as in IHC.

Authors: We agree with the reviewer. We have now redesigned this part of Fig.7 as we have performed more in vivo experiments in the meantime and importantly now also analysed endpoint results (day 14). In particular, we show now that TPC2 KO and Rab7 KO in vivo result in significant reduction of tumour weight compared to WT, in line with our in vitro findings on tumour hallmarks after either TPC2 or Rab7 KO/KD. In addition, the reduced tumour weight after Rab7 KO cell injection could be partially reversed with TPC2-A1-P agonist treatment, reaching values that were not significantly different from WT at 14 days. This is in line with the bioluminescence data showing significantly increased bioluminescence signal after TPC2-A1-P treatment in Rab7 KO versus vehicle treated Rab7 KO control already at day 7 (Fig.7n). By contrast, application of a Rab7 agonist, ML-098 did have no effect on tumour weight in TPC2 KO, further supporting the claim from our in vitro (rescue) experiments that TPC2 OE or activation can reverse the effect in Rab7 KO while Rab7 OE or activation cannot rescue TPC2

KO. Specifically, we now show the isolated tumours at day 14 when the mice were sacrificed (new Fig.7m-q).

We now also provide additional data on migration behavior in vivo. We performed a tumour dissemination assay in which tumour cells are allowed to migrate to distant organs after tail vein injection. In accordance with our reported in vitro data (see also new Suppl. Fig.S7h-I and below) and in line with previous publications on the impact of TPC2 KO in cancer cell migration (Müller et al., Cell Chem Biol., 2021, Nguyen et al., Cancer Res., 2017), dissemination of cancer cells is impaired upon knock-out of TPC2 or Rab7. Unfortunately, we have no approval for testing the compounds, TPC2-A1-P and Rab7 agonist ML-098 in this model. A de novo application with the Bavarian Government would take >12 months.

Minor Comments:

1. Direct Effect of Rab7a on TPC2: This study did not demonstrate a direct interaction between Rab7a and TPC2.

Authors: In Point 4 the reviewer acknowledges that the interaction between Rab7 and TPC2 had been shown before (the reviewer refers here specifically to Lin-Moshier et al., PNAS, 2016). We have confirmed this interaction by Co-IP experiments but beyond that now also provide FRET experiments and importantly patch-clamp experiments, thus demonstrating not only physical but also functional interaction of the two proteins. Of note, Lin-Moshier et al. had already mapped the binding site to the N-terminus of TPC2 (residues 33-37), further supporting direct interaction.

We believe that the amount of new data (e.g., the FRET data, the rapid effect of Rab7 inhibitor and the GTP effect) combined with the previous data regarding physical interaction (CoIP and mapping of the binding site), attributes to a substantial body of evidence to support the claim of direct physical and functional interaction. See also Point 1 (Major) above.

2. Molecular Mechanism: Using a small molecule inhibitor of Rab7 is insufficient to elucidate the molecular mechanism.

Authors: We thank the reviewer for their comment. The aim of the Rab7 inhibitor experiment was to further corroborate the regulatory effect of Rab7 on TPC2 activity and to also elucidate how quickly this effect can be seen on a functional level. We found in electrophysiology experiments that Rab7 small molecule inhibition shows similar results on TPC2 activity as Rab7KO and that the effect of Rab7 on TPC2 is indeed instant and very fast, supporting the claim of a direct effect rather than being e.g. a consequence of changes in the transcriptome or proteome. Please see also Point 2 (Major) above on mechanism.

3. mRNA vs. Protein: Clear differentiation between mRNA and protein is necessary in describing the text.

Authors: We have changed this accordingly by mentioning “qPCR” or “mRNA” where “expression” refers to qPCR.

4. Previous Reports: The Rab7-TPC2 interaction was previously reported in PNAS 2016.

Authors: See also Point 4 (Major) and Point 1 (Minor) above.

In addition, we would like to highlight that building upon the finding of interactomes in the Lin-Moshier et al., PNAS, 2016 paper, we have further substantiated the functional significance of this interaction by GCaMP and electrophysiology experiments but also pathophysiologically in melanoma. Our investigations have demonstrated that knockout, knockdown, or inhibition of Rab7a or TPC2, contributes to melanoma hallmarks and tumorigenesis in vitro and in vivo. To reinforce this observation and in order to emphasize the clinical relevance of our findings, we have conducted tissue staining analyses comparing lymphnode sections from healthy human individuals to metastatic melanoma patients. Our data (provided below) revealed increased expression of Rab7 and MITF in the majority of melanoma tissues compared to healthy controls. This underscores the role of Rab7 as a

possible melanoma driver, increasing melanoma hallmarks by enhancing the activity of TPC2. See new Fig.1 and 6. In sum, we believe our study goes way beyond current knowledge, in particular with regards to the functional interaction between Rab7a and TPC2 and their impact on MITF.

5. Endogenous Interaction: Was the endogenous interaction between Rab7a and TPC2 demonstrated?

Authors: See Point 6. We show functional interaction between Rab7a and TPC2 endogenously in Fig.3 (Electrophysiology in TPC2 and Rab7 KO SKMEL5 cells). This is a very important experiment showing not only that our OE data are reproducible in the endogenous system but also that this functional interaction is relevant in endogenously expressing melanoma cells.

6. Experimental Design: Justification is needed for not using melanocytes or melanoma in electrophysiology experiments.

Authors: We have used melanoma cell lines (i.e. SK-MEL-5) in our electrophysiology studies as well as for our pathophysiological studies for several reasons: high expression levels of both TPC2 and Rab7 and successful production of CRISPR-Cas9 engineered KOs for TPC2 and Rab7. The availability of these KOs was crucial for the study to demonstrate that TPC2 channel activity was reduced in Rab7KO (and absent in TPC2KO as control, and to demonstrate specificity of our compound), thus validating our findings in the overexpression system. Please see Fig.3 and the following section in the MS text:

„The electrophysiological analysis revealed that the activity of TPC2 was absent in TPC2 KO and significantly reduced in Rab7a KO compared to WT SK-MEL-5 cells (Fig. 3k-l), corroborating the results obtained from HEK293 presented in Fig. 2.“

With melanoma samples from human patients we could not have done this (generation of CRISPR/Cas9 KOs in primary tissue/cells is very challenging). With primary melanocytes this is

likewise challenging and we wanted to investigate Rab7 and TPC2 KO in disease/cancer models and not in healthy cells. Nevertheless, we have tried to patch-clamp lysosomes in human primary melanocytes. Unfortunately, we had difficulties obtaining a sufficient number of enlarged lysosomes as well as sufficiently enlarged lysosomes in these cells. We tried a plethora of tools such as vacuolin, apilimod, YM201636 without much success. Either lysosomes were only marginally enlarged or trying to isolate lysosomes which seemed big enough for patch clamp failed because the cells detached very quickly or no stable seals could be achieved. This is unfortunate since we have managed so far to patch clamp lysosomes from a variety of cells such HEK, HeLa, SK-MEL-5 but also from e.g. fibroblasts, macrophages, mast cells, or cardiomyocytes.

However, we have performed endolysosomal patch-clamp experiments not only in melanoma lines but also in other, non-melanoma cancer lines for comparison i.e., HeLa (cervical cancer) and MDA-MB-231 (breast cancer) to corroborate the finding that the latter ones express lower levels of TPC2 and Rab7, respectively as suggested by the qPCR and WB data. See Suppl. Fig.S1. In line with the expression data, the electrophysiological analysis of the other cancer lines revealed significantly lower TPC2 activities than in the melanoma cells.

In addition, we do provide now more evidence from human melanoma patient and healthy control samples as outlined above. Expression analysis of these samples revealed elevated Rab7 levels in the majority of melanoma tissues compared to controls. See also Point 4 (Minor).

7. Fig. 6. The MITF story is another biased stretch. Again, correlation does not guarantee causation. In other words, picking MITF over many other proteins or molecules involved in melanoma tumorigenesis and WNT signaling was not justified or convincing. Descriptive results are insufficient to demonstrate that TPC2-RAB7A's impact is mediated by MITF without rescue results (using MITF WT as well as mutants [if available]).

Authors: We have selected MITF due to its close association with the endolysosomal machinery. Notably, MITF transcript levels not only exhibit a significant correlation with lysosomal genes in melanoma but MITF also actively promotes the transcription of such genes in melanoma cells (see e.g., Ploper et al., PNAS, 2015). We have also illustrated the correlation of Rab7 and MITF in the cell lines utilized in this study (Figures 6a and b), wherein cell lines exhibiting high Rab7 protein levels also display elevated MITF protein levels. That MITF correlates with Rab7 has been shown before in microarray data as well:

<https://www.sciencedirect.com/science/article/pii/S1043661815000687>

In addition, we have now examined also other pathways and other components of the Wnt signaling pathway. See Point 8 below and Point 1 above.

8. WNT Signaling: WNT signaling is frequently hyperactivated in melanoma. Adding the following to the Discussion is highly recommended (only if the authors want to include the WNT signaling in this manuscript): Why is an additional layer of regulation (positive) of the WNT signaling necessary for melanoma tumorigenesis? In other words, is it also possible that the impact of the TPC2-RAB7A axis on melanoma tumorigenesis is WNT signaling-independent? What is the rationale for the authors to consider only WNT signaling to explain the TPC2-RAB7A axis in melanoma tumorigenesis?

Authors: We agree that this is an important point. The Wnt/GSK3 β / β -Catenin/MITF pathway is closely linked to the endolysosomal system (see e.g., Ploper et al., PNAS, 2015 or <https://pubmed.ncbi.nlm.nih.gov/21183076/>), and its functionality relies on a properly working endolysosomal machinery and trafficking system.

Our data revealed a reduction of MITF levels in TPC2 and Rab7KO melanoma lines. Several signaling pathways can impact MITF degradation and/or MITF expression/transcription such as Wnt/GSK3 β / β -Catenin, cKit/RAS/RAF/MEK/ERK, cKit/RAS/PI3K/Akt or α MSH/cAMP/CREB (<https://pubmed.ncbi.nlm.nih.gov/18927540/>). Importantly, there is also crosstalk between pathways e.g., Akt inhibits GSK3 β .

We only want to claim here that specific Wnt signaling pathway components such as GSK3 β or β -Catenin are affected by Rab7a/TPC2 KO/KD, not necessarily the entire signaling cascade or even membrane receptors (Wnt/Frizzled). We claim that, when TPC2 or Rab7a are absent, GSK3 β along with other components of the destruction complex is not sequestered properly into late endosomes/lysosomes, resulting in elevated GSK3 β (confirmed by exp.) as well as decreased β -Catenin (confirmed by exp.). GSK3 β phosphorylates β -catenin, rendering it unstable and unable to translocate to the nucleus, thereby decreasing its expression level, resulting in reduced MITF levels due to increased proteasomal degradation and reduced β -Catenin mediated transcription. To further back up our claims we investigated the effect of Rab7 on GSK3 β in Rab7 and TPC2KOs as outlined above. In sum, our findings corroborate the hypothesis that Rab7a acts as an effector of TPC2 but not vice versa and that both Rab7 and TPC2 KO show a strong regulatory effect on central Wnt-signaling components such as GSK3 β and β -catenin. We could also demonstrate that OE of GSK3 β further reduces proliferation of Rab7a KO as well as TPC2 KO SK-MEL-5 cells (new Fig. S6c). See also Points 1 and 3 (Major) above.

As for the signaling cascades, we have now examined other pathways in addition to the Wnt/ β -Catenin pathway upstream of MITF. This panel encompassed PI3K/Akt and cAMP/CREB pathway components (Fig. S9). While pCREB/CREB ratios were not significantly changed in the KOs vs. WT, pAkt/Akt ratios were increased in both KOs, which theoretically should result in increased MITF levels. However, since in both Rab7a and TPC2 KO lines we found reduced MITF levels, the increase in Akt likely reflects a compensatory effect in response to the changes in the Wnt-signaling pathway and the degradation of MITF due to TPC2 or Rab7a KO.

In sum, we think that our data sufficiently support the claim that TPC2/Rab7a KO/KD affects melanoma tumorigenesis via regulation of Wnt signaling components.

Reviewer #4 (Remarks to the Author):

An interesting study by Abrahamian et al. about the role of small GTPase RAB7A as an oncogene in melanoma cancer, in vitro and in vivo. The work was **well written**, and I appreciated how the experiments were designed and how each finding fueled the next question. **I recommend this paper be published**, however there are several major concerns in this work as shown in following comments:

Authors: We appreciate the reviewer's support, their comments that the study is well written and designed and that they recommend the paper for publication. We have addressed the reviewer's remaining concerns as outlined below.

- The authors declared that RAB7A is an effector of TPC2, but they did not demonstrate the effect on endosomal function. I suggest to perform in vitro and ex vivo experiments to observe the effect of TPC2 KO/KD and RAB7A KO/KD on the lysosomal pathway, acidification, and function (immunofluorescence for lysosomal activity, size, and number; valuation of expression levels of endosomal trafficking proteins; analysis with TEM microscopy of endosomal compartments). Indeed, to the best of my knowledge, the KO of RAB7A is lethal in vivo. Thus, it is plausible that in the cell lines, RAB7 KO induces compensatory effects and other pathway activation.

Authors: We agree with the Reviewer that either loss of TPC2 or Rab7a strongly affects lysosomal function e.g., endolysosomal trafficking and degradation, lysosomal activity etc. This has been shown extensively in previous publications for both TPC2 and Rab7a. Even more striking both Rab7a and TPC2 KO often result in similar effects, further supporting close cooperativity of the two proteins. We have extensively alluded to that in our Introduction in the MS:

„Rab7a plays a fundamental role not only for trafficking and degradation of many signaling receptors e.g., EGF/EGFR and adhesion molecules, but also for biogenesis, positioning, and motility of lysosomes as well as auto- and phagolysosomes¹⁻⁴. Rab7a further plays key roles in cell survival, growth, differentiation, migration, autophagy and apoptosis. Modulation of Rab7a activity affects a number of disease pathologies including neuropathies and neurodegenerative diseases such as Charcot-Marie-Tooth type 2B, hereditary sensory neuropathy type 1, and Niemann Pick type C1 (NPC1), infectious diseases, and cancer, including melanoma⁵⁻¹⁰. Thus, Rab7a is e.g., associated with poor prognosis of gastric cancer and promotes proliferation, invasion, and migration of gastric cancer cells. Knockdown of Rab7a suppresses the proliferation, migration, and xenograft tumor growth of breast cancer cells and high Rab7a expression is an indicator of a higher risk of metastasis in early melanoma patients. Furthermore, in melanoma cells Rab7a levels are significantly elevated compared to normal skin melanocytes, impacting melanoma proliferation and invasion^{5,9,11,12}. Similar to Rab7a, the Na⁺ and Ca²⁺ permeable cation channel TPC2 in LE/LY and melanosomes of melanocytes has been described as an important regulator of endolysosomal trafficking, with EGF/EGFR, LDL cholesterol, or PDGF accumulating in TPC2 knockout cells¹³⁻¹⁶. In analogy to TPC2 inhibition, knockdown or knockout, Rab7a knockdown in NPC1 cells exacerbates cholesterol accumulation^{5,17,18}. Inhibition, knockdown or knockout of TPC2 also results in reduced proliferation, migration, and invasion as well as reduced tumour growth, metastasis

formation and VEGF-induced angiogenesis in different types of cancer, including melanoma^{16,19-23}.

The role of TPC2 and Rab7 in lysosomal function has been extensively studied and validated over many years. The novelty of our study is that TPC2 and Rab7 not only show similar functions but that they functionally interact, demonstrated by patch-clamp electrophysiology and GCaMP based Ca²⁺ imaging experiments. This cooperativity results in similar effects in cancer cells such as melanoma cells, specifically with a striking effect on melanoma cells that express high levels of both proteins. We show increased levels of GSK3 β in Rab7 and TPC2KO, in line with the reduced degradation of GSK3 β in endolysosomes with the consequence that MITF proteasomal degradation is hampered as shown by Ploper et al., PNAS, 2015. Hence, defects in endolysosomal function as previously shown for Rab7 or TPC2KO would be expected to be disrupted in our KO cells.

To illustrate this and to specifically address the reviewer's comment, we show now data for Rab7KO and TPC2KO on endolysosomal function in the melanoma cells we used in this study. Specifically, we analysed endolysosomal size and number using Lysotracker and we probed endolysosomal trafficking using dextran-FITC. We also assessed endolysosomal exocytosis capability (Fig. S8). In sum these data confirm previously published data on endolysosomal defects in Rab7 as well as TPC2 KO cells.

• Alfonso-Curbelo declared that, in the same cell lines used in this work, regulation of RAB7A is MITF-independent demonstrating that levels of RAB7 are different in the progression of melanoma. In particular, he demonstrated that the oncogenic transcription factor Myc is activated and induces a strong overexpression of Rab7. How do the authors justify MITF dependence in the same cell lines? I think that the authors should demonstrate that silencing MITF induces RAB7 and TPC2 downregulation and vice versa. Moreover, to validate their hypothesis they should demonstrate the increase of GSK3B degradation when RAB7A and TPC2 are upregulated and vice versa. Indeed, the authors "postulated" this mechanism, but they do not demonstrate it.

Authors: We have shown in our study that Rab7 or TPC2 KO/KD reduce MITF and increase GSK3 β . While Dr. Soengas (who is also co-author on this MS) and colleagues did not observe significant changes in Rab7 expression upon knockdown of MITF, it is noteworthy that experiments exploring the reverse effect i.e., the effect of Rab7 KD/KO on MITF, have not been performed by these authors. One of the conclusions drawn by Alfonso-Curbelo et al. was that the “dependency of melanoma cells on Rab7 is independent of [...] basal MITF levels”. This statement contrasts the data published by Ploper et al., PNAS, 2015: “MITF expression in the tetracycline-inducible C32 melanoma model caused a marked increase in vesicular structures, and increased expression of late endosomal proteins, such as Rab7, LAMP1, and CD63.” That MITF correlates with Rab7 has also been confirmed by microarray data:

<https://www.sciencedirect.com/science/article/pii/S1043661815000687>

We postulate here that the MITF levels are dependent on Rab7 and TPC2 activity, hence we looked at the reverse effect. Thus, our data demonstrate that reduced TPC2/Rab7 activity, resulting in increased GSK3 β levels, reduces MITF levels.

To add to that we have now performed additional experiments using our CRISPR/Cas9 engineered SK-MEL-5 Rab7-KO and TPC2-KO lines as the reviewer requested. While the KO of Rab7 and TPC2 have shown increased GSK3 β expression (Fig.6f-g and h-i), overexpressing either Rab7aWT or Rab7aQ67L in Rab7 KO shows rescue, i.e. decrease of GSK3 β expression. Importantly and in line with our proliferation and invasion data, the overexpression of Rab7a in TPC2 KO did not result in any differences in GSK3 β protein levels, further confirming that Rab7a overexpression cannot rescue in the absence of TPC2 (new Suppl. Fig.6a,b and below).

In addition, we now also demonstrate that OE of GSK3 β further reduces proliferation of Rab7a KO as well as TPC2 KO SK-MEL-5 cells (new Fig. S6c and below).

We have added a section on this matter in the Results and Discussion and also highlighted in the revised MS several relevant points that are in line between the two studies:

„In accordance with our findings, Alonso-Curbelo et al. (2014) postulated that Rab7 is an early induced melanoma driver and found reduced proliferation after Rab7 shRNA treatment for different melanoma lines e.g., UACC-62 or WM-164 while Rab7 shRNA treatment was much less efficient in reducing proliferation of SK-MEL-103 cells, becoming significant in the latter only after 4 days. After 3 days instead the effect was not significant, in line with our results⁹. Likewise, non-melanoma cancer lines were barely affected by Rab7 shRNA treatment. Alonso-Curbelo et al. (2014) further found reduced tumour volume after Rab7 shRNA treatment, however again with much more prominent effects in UACC-62 (high MITF) as compared to SK-MEL-103 or SK-MEL-147 (low MITF).“

Minor: The authors compared the expression of RAB7A to the expression of RAB7B. I think that this comparison is useless because these two proteins are not isoforms, their identity is limited to 50% and they do not share the RabF and RabFS motifs. Indeed, these proteins have also very different functions, have differential localization and control different steps of transport. RAB7B is involved in the control of endosomes to Golgi transport, being localized both to the Trans Golgi Network and late endosomes.

Authors: We agree with the reviewer that Rab7a and b have very different functions. Nevertheless, we thought it might be interesting to see if Rab7b may also be expressed at high levels in melanoma and if yes whether it might also interact and/or affect TPC2 activity; just like we have also explored whether TRPML1 and Rab7a functionally interact albeit TPC2 and TRPML1 are also unrelated, distinct and very different proteins. Illustrating the low expression of Rab7b in melanoma and non-melanoma cells, helps IOO to mitigate any potential compensatory effects attributed to Rab7b and to provide more confidence in assigning the observed effects solely to Rab7a.

We would argue therefore to not omit these data from the MS as they add additional (potential useful) information for future readers. Of course, if the reviewer wants them removed we will remove them.

REVIEWER COMMENTS

Reviewer #1 (Remarks to the Author):

The authors have done a lot of extra work to address the concerns of reviewers and I now believe that the manuscript should be published.

Reviewer #2 (Remarks to the Author):

The reviewers have addressed my concerns in the revisions.

Reviewer #3, Withdrawn

Reviewer #4 (Remarks to the Author):

Thank to the authors. They have revised the manuscript according to my comments.

Reviewer #5 (Remarks to the Author):

For this second round review, I was asked to specifically evaluate whether the authors had addressed the comments made by reviewer #3, with a focus on WNT signaling. This is what I have done. I will leave any other comments on the Rab7/TPC2 interaction and electrophysiology and FRET experiments to others.

1. One major concern expressed by this reviewer was that the link with WNT signaling was weak to non-existent. As the authors themselves remark in the rebuttal, they only “want to claim that Wnt signaling pathway components such as GSK3 β and β -Catenin are affected by Rab7a/TPC2 KO/KD, not necessarily the entire signaling cascade and/or even receptor signaling”.

I agree with both the original reviewer 3 and the authors. For this reason, however, I would recommend that the authors modify the title to “Rab7a is a direct effector of the intracellular Ca²⁺ channel TPC2 regulating melanoma progression through modulation of GSK3 β ”.

As far as I can tell, this is the strongest claim the authors can make - maybe they can include “GSK3 β and MITF”. Even then it is a bit of a stretch, because the only link the authors show for GSK3 is in vitro proliferation, there are no in vivo data also linking the proliferation and invasion to either GSK3 β , CTNNB1 or MITF - but that doesn't have to be a problem if the authors are very clear about the strength of their evidence throughout. Here, I feel they do drop the ball.

2. If the authors focus on GSK3 β and CTNNB1, they cannot talk about “the WNT pathway” - this also implies signaling activity through CTNNB1, which they do not test. At the very least they would require an AXIN2 qRT-PCR or a TCF/LEF reporter assay).

They also do not show that blocking the WNT pathway abrogates the effects of TPC2 and Rab7A, nor do they show that the MITF effect has anything to do with CTNNB1. GSK3 is a

multi-tasking kinase, so without investigating the WNT pathway involvement and links in more detail as mentioned above, they cannot claim that Rab7A acts through modulation of the WNT pathway. The current title is simply misleading.

3. I appreciate the authors' efforts to also show involvement of CSNK1a (and I understand the difficulty of showing protein level changes for Axin and APC, which is technically challenging if not impossible with current antibody tools), but this again is a multitasking kinase which is not entirely specific for the WNT pathway. Again, the most convincing proof for WNT pathway involvement is changes in de-phosphorylated CTNNB1 and/or Axin2 mRNA levels. In the absence of both, the authors need to town down their claim.

4. Regarding major comment 3 raised by reviewer 3:

I don't think the reviewer states that the Rab7/TPC2 is not causal for the observed proliferation/invasion effects, but that the involvement of GSK3 and/or MITF is not proven to be the causal mechanism by the authors. They authors talk around this issue in their rebuttal (reviewer 4 also asks for such an experiment, and they get a similar evasive answer). Thus, reviewer 3's point still stands and I agree with them. The MITF involvement is suggestive, but not demonstrated: For this the authors would have to modify (increase or decrease MITF levels) and show that they lose the effect on proliferation/invasion (or similar for GSK3 if they want to make that their focus).

Of course it is OK to speculate, but this should be done carefully so as not to muddle the scientific record.

5. As for the involvement of other pathways: The differences in p-Akt (new FigS9b) are much more prominent than those in GSK3b - still the authors prefer GSK3 as the responsible driving mechanism. Which is a fair choice, but nevertheless noticeable.

6. Minor comment 8 of reviewer 3 regarding WNT signaling: The authors do provide an elaborate response in the rebuttal, but don't address this issue in the actual text of the manuscript. They could have expanded the discussion to include a balanced discussion on the role of CTNNB1-dependent (WNT3A-driven) and -independent (WNT5A-driven) WNT signaling in melanoma and how this is know to control a switch between proliferation and invasion, but they don't. We don't know what happens to GSK3, CTNNB1 or MITF in the in vivo experiments.

7. Altogether, the authors seem set on making the claim that the WNT pathway is involved without understanding or addressing the intricacies at play. I suggest they tone down the title and claims in the paper - it's fine if they speculate in the discussion as to the potential mechanism, but more than that would be overinterpreting their own data.

Specifically, lines 234-240 make a direct link between WNT signaling and MITF, essentially describing a situation in which MITF is akin to CTNNB1 and stabilized upon WNT signaling. Similarly, lines 254-255 on destruction complex involvement seem a bit of a stretch.

Also, lines 354-355 on MITF being a downstream (presumably transcriptional?) target of CTNNB1 would mean an additional (or different) mechanisms is at play.

Again: it is OK to speculate, but the authors currently connect different elements of the literature into a presumed mechanism that remains ultimately unclear. Can they at the very least provide a model for how they think all of this works?

My suggestion would be that the authors make a much more straightforward link to the

Proper study (their ref. 30) which shows that MITF contains GSK3 phosphorylation sites. That link seems clear, and it's OK to use that as an entry to follow up on GSK3 and MITF, but this can be done without drawing in the WNT pathway. In fact, the evidence for a direct change in MITF protein levels upon a WNT signal is not all that clear in the original Proper study, as far as I see. Neither is the order and/or involvement of GSK3B/CTNNB1/MITF evident.

8. A very simple experiment the authors could do, is to take their control and Rab7KO (and if needed their TPC2 KO cells), treat those with and without WNT3A and test if the CTNNB1 and MITF response is different between the two (and with response I then mean CTNNB1 and MITF protein levels and ideally downstream WNT signaling in the form of an Axin2 qPCR or TCF/LEF luciferase reporter assay. This would also tell them if the overall lower levels of CTNNB1 that they see in Figure 6F are also functionally meaningful.

9. FigS6 also lacks the accompanying CTNNB1 and MITF blots: is their expression also restored to wildtype as the GSK3 levels are restored? This seems like an obvious thing to have included. Again, in the absence of such data the authors can only claim mechanistic involvement of GSK3b.

10. The methods, by the way, in lines 580-582 lack crucial details on primary antibodies used so I cannot comment on the quality of the blots. For CTNNB1 the authors would be advised to not only use a total CTNNB1 antibody but also a non-phosphorylated CTNNB1 antibody to measure changes in active CTNNB1 signaling if that's the route they want to go down.

REVIEWER COMMENTS

Reviewer #1 (Remarks to the Author):

The authors have done a lot of extra work to address the concerns of reviewers and I now believe that the manuscript should be published.

Reviewer #2 (Remarks to the Author)

The reviewers have addressed my concerns in the revisions.

Reviewer #3, Withdrawn

Reviewer #4 (Remarks to the Author):

Thanks to the authors. They have revised the manuscript according to my comments.

Authors: We would like to thank the 3 Reviewers for their efforts and for their positive evaluations. We are happy to hear that all 3 Reviewers are fully satisfied with our efforts to improve the MS.

Reviewer #5 (Remarks to the Author):

For this second round review, I was asked to specifically evaluate whether the authors had addressed the comments made by reviewer #3, with a focus on WNT signaling. This is what I have done. I will leave any other comments on the Rab7/TPC2 interaction and electrophysiology and FRET experiments to others.

1. One major concern expressed by this reviewer was that the link with WNT signaling was weak to non-existent. As the authors themselves remark in the rebuttal, they only “want to claim that Wnt signaling pathway components such as GSK3 β and β -Catenin are affected by Rab7a/TPC2 KO/KD, not necessarily the entire signaling cascade and/or even receptor signaling”.

I agree with both the original reviewer 3 and the authors. For this reason, however, I would recommend that the authors modify the title to “Rab7a is a direct effector of the intracellular Ca²⁺ channel TPC2 regulating melanoma progression through modulation of GSK3beta”.

As far as I can tell, this is the strongest claim the authors can make - maybe they can include "GSK3b and MITF". Even then it is a bit of a stretch, because the only link the authors show for GSK3 is in vitro proliferation, there are no in vivo data also linking the proliferation and invasion to either GSK3B, CTNNB1 or MITF - but that doesn't have to be a problem if the authors are very clear about the strength of their evidence throughout. Here, I feel they do drop the ball.

2. If the authors focus on GSK3B and CTNNB1, they cannot talk about "the WNT pathway" - this also implies signaling activity through CTNNB1, which they do not test. At the very least they would require an **AXIN2 qRT-PCR or a TCF/LEF reporter assay**).

They also do not show that blocking the WNT pathway abrogates the effects of TPC2 and Rab7A, nor do they show that the MITF effect has anything to do with CTNNB1. GSK3 is a multi-tasking kinase, so without investigating the WNT pathway involvement and links in more detail as mentioned above, they cannot claim that Rab7A acts through modulation of the WNT pathway. The current title is simply misleading.

Authors (Points 1 and 2): We agree with the Reviewer and rather than merely modifying the title and the claims (which we nevertheless did to conform with the reviewer's request) we decided to provide also additional experimental data to further buttress our claims. See below.

3. I appreciate the authors' efforts to also show involvement of CSNK1a (and I understand the difficulty of showing protein level changes for Axin and APC, which is technically challenging if not impossible with current antibody tools), but this again is a multitasking kinase which is not entirely specific for the WNT pathway. **Again, the most convincing proof for WNT pathway involvement is changes in de-phosphorylated CTNNB1 and/or Axin2 mRNA levels. In the absence of both, the authors need to town down their claim.**

Authors: We agree with the Reviewer and have now performed phospho-betaCatenin experiments.

GSK3beta phosphorylates beta-Catenin for degradation:

“Under physiologic conditions, β -catenin levels are tightly regulated by the β -catenin destruction complex. The assembly of this complex is mediated by a central scaffold protein, axin, which has binding sites for β -catenin and the other complex components adenomatous polyposis coli (APC), casein kinase 1 α (CK1 α), and glycogen synthase kinase 3 (GSK3). Within this complex, β -catenin is phosphorylated earmarking it for subsequent ubiquitination and proteasomal degradation.”

<https://www.nature.com/articles/s41467-019-12203-8>

Accordingly, phospho-betaCatenin levels would be expected to be increased in the KOs.

We have performed WB experiments now and found that phospho-betaCatenin was indeed increased.

4. Regarding major comment 3 raised by reviewer 3: I don't think the reviewer states that the Rab7/TPC2 is not causal for the observed proliferation/invasion effects, but that the involvement of GSK3 and/or MITF is not proven to be the causal mechanism by the authors. They authors talk around this issue in their rebuttal (reviewer 4 also asks for such an experiment, and they get a similar evasive answer). Thus, reviewer 3's point still stands and I agree with them. The MITF involvement is suggestive, but not demonstrated: For this the authors would have to modify (increase or decrease MITF levels) and show that they lose the effect on proliferation/invasion (or similar for GSK3 if they

want to make that their focus). Of course it is OK to speculate, but this should be done carefully so as not to muddle the scientific record.

Authors: We have presented previously data showing that increased GSK3 β levels in Rab7a KO could be reduced again when overexpressing either Rab7aWT or Rab7aQ67L (Fig. S6a). Vice versa, increased GSK3 β levels in TPC2 KO could not be reversed by overexpressing either Rab7aWT or Rab7aQ67L (Fig. S6b), in line with a role of TPC2 as downstream target of Rab7a. In addition, we showed that overexpression of GSK3 β was able to further reduce proliferation in both Rab7a and TPC2 KO SK-MEL-5 lines (Fig. S6c).

Similar experiments have now been performed for β -Catenin (Fig. S6d-e; rescue data) and MITF (proliferation data). In the latter experiments we checked for proliferation after MITF OE in WT, Rab7 or TPC2 KO cells. Importantly, overexpression of MITF was able to increase proliferation in WT, confirming a direct effect of MITF on melanoma cell proliferation, while both Rab7a and TPC2 KO SK-MEL-5 lines did not show a difference in proliferation compared to empty vector expression, likely because MITF OE in the KOs can still not keep up with the continued degradation.

See also Point 9. In addition, we show now new ex vivo experiments demonstrating that MITF and β -Catenin are strongly reduced in the isolated tumours in case of TPC2 KO and Rab7a KO, further confirming that reduction of MITF and β -Catenin is also relevant in vivo.

5. As for the involvement of other pathways: The differences in p-Akt (new FigS9b) are much more prominent than those in GSK3b - still the authors prefer GSK3 as the responsible driving mechanism. Which is a fair choice, but nevertheless noticeable.

Authors: Indeed, we found that pAkt/Akt ratios were increased in the KOs, which theoretically should result in increased MITF levels (the cartoon below by Hocker et al. illustrates this) as Akt block on GSKbeta would lead to MORE beta-Catenin and thus more MITF. However, in both Rab7a and TPC2 KO lines we found reduced MITF levels. Hence, we concluded that the Akt effects we see cannot be causative for the reduced MITF levels but may “reflect a compensatory effect in response to the changes in the Wnt-signaling pathway and the degradation of MITF due to TPC2 or Rab7a KO.” We hope this makes it clearer now.

[Redacted]

Hocker et al., J Invest Dermatol 128:2575-2595, 2008

6. Minor comment 8 of reviewer 3 regarding WNT signaling: The authors do provide an elaborate response in the rebuttal, but don't address this issue in the actual text of the manuscript. They could have expanded the discussion to include a balanced discussion on the role of CTNNB1-dependent (WNT3A-driven) and -independent (WNT5A-driven) WNT signaling in melanoma and how this is known to control a switch between proliferation and invasion, but they don't. We don't know what happens to GSK3, CTNNB1 or MITF in the in vivo experiments.

Authors: We agree with the Reviewer and have now expanded our Discussion to include a balanced discussion on the role of CTNNB1 in WNT signaling in cancer/melanoma (marked in red in the new MS text).

7. Altogether, the authors seem set on making the claim that the WNT pathway is involved without understanding or addressing the intricacies at play. I suggest they tone down the title and claims in the paper - it's fine if they speculate in the discussion as to the potential mechanism, but more than that would be overinterpreting their own data.

Specifically, lines 234-240 make a direct link between WNT signaling and MITF, essentially describing a situation in which MITF is akin to CTNNB1 and stabilized upon WNT signaling.

Similarly, lines 254-255 on destruction complex involvement seem a bit of a stretch.

Authors: These statements are based on literature results, which we have cited accordingly. All we do here is repeating what has been claimed in the works that we cite, e.g. Ploper et al., PNAS, 2015:

"A custom-made antiphospho antibody confirmed that the novel C-terminal sites on MITF were indeed phosphorylated by GSK3. The results suggest a positive regulatory loop by which MITF expands MVBs/late endosomes, resulting in increased Wnt signaling which in turn stabilizes MITF by decreasing its GSK3 phosphorylations."

or

"The results indicated that the destruction complex components Axin1, GSK3, and p-β-catenin are translocated to CD63+ endolysosome/MVBs induced by MITF in the presence of Wnt. The enhanced sequestration of GSK3 and Axin1 provides a cell biological explanation for how MITF induction increased Wnt signaling (Fig. 4F)."

or

"Without Wnt signaling, GSK3 phosphorylates MITF on novel C-terminal phosphorylation sites, targeting MITF for proteasomal degradation. Upon Wnt signaling, destruction complex components are sequestered into MVBs, inhibiting GSK3 and stabilizing MITF. In turn, MITF induces late endolysosomes that further sequester destruction complex components upon Wnt signaling, enhancing overall Wnt responsiveness. This positive-feedback loop is proposed to function in the proliferative stages of melanoma, in which MITF and Wnt signaling peak."

Nevertheless, we write now "reportedly" at several occasions to tone down our statements.

Also, lines 354-355 on MITF being a downstream (presumably transcriptional?) target of CTNNB1 would mean an additional (or different) mechanism is at play. Again: it is OK to speculate, but the authors currently connect different elements of the literature into a presumed mechanism that remains ultimately unclear. Can they at the very least provide a model for how they think all of this works?

Authors: Our claims and conclusions are all based on evidence from the literature, see e.g. cartoon below from the paper by Ploper et al., PNAS, 2015 or the paper/cartoon above by Hocker et al., J Invest Dermatol 128:2575-2595, 2008.

[Redacted]

Legend to this Fig. from Ploper et al., PNAS, 2015: “Without Wnt signaling, GSK3 phosphorylates MITF on novel C-terminal phosphorylation sites, targeting MITF for proteasomal degradation. Upon Wnt signaling, destruction complex components are sequestered into MVBs, inhibiting GSK3 and stabilizing MITF. In turn, MITF induces late endolysosomes that further sequester destruction complex components upon Wnt signaling, enhancing overall Wnt responsiveness. This positive-feedback loop is proposed to function in the proliferative stages of melanoma, in which MITF and Wnt signaling peak.”

We have now changed the title and toned down statements and claims relating to Wnt signaling in the MS. In addition as requested, we show now a summary cartoon to provide a model for how we think this may all work, see below.

My suggestion would be that the authors make a much more straightforward link to the Ploper study (their ref. 30) which shows that MITF contains GSK3 phosphorylation sites. That link seems clear, and it's OK to use that as an entry to follow up on GSK3 and MITF, but this can be done without drawing in the WNT pathway. In fact, the evidence for a direct change in MITF protein levels upon a WNT signal is not all that clear in the original Ploper study, as far as I see. Neither is the order and/or involvement of GSK3B/CTNNB1/MITF evident.

Authors: This is also supported by other literature as summarized in the cartoon above by Hocker et al., J Invest Dermatol 128:2575-2595, 2008

Based on literature evidence as well as supported by our new data we would postulate the mechanism as shown in the cartoon below. We have added this cartoon as Fig.8 in the MS now.

8. A very simple experiment the authors could do, is to take their control and Rab7KO (and if needed their TPC2 KO cells), treat those with and without WNT3A and test if the CTNNB1 and MITF response is different between the two (and with response I then mean CTNNB1 and MITF protein levels and ideally downstream WNT signaling in the form of an Axin2 qPCR or TCF/LEF luciferase reporter assay. This would also tell them if the overall lower levels of CTNNB1 that they see in Figure 6F are also functionally meaningful.

Authors: We agree with the Reviewer. See Point 3. We have performed now additional experiments as requested to further back up our claims such as the suggested phospho-betaCatenin WB.

9. FigS6 also lacks the accompanying CTNNB1 and MITF blots: is their expression also restored to wildtype as the GSK3 levels are restored? This seems like an obvious thing to have included. Again, in the absence of such data the authors can only claim mechanistic involvement of GSK3b.

Authors: We agree with the Reviewer and now provide such rescue exp. for betaCatenin. See also new Fig.S6d-e. and Point 4.

10. The methods, by the way, in lines 580-582 lack crucial details on primary antibodies used so I cannot comment on the quality of the blots. For CTNNB1 the authors would be advised to not only use a total CTNNB1 antibody but also a non-phosphorylated CTNNB1 antibody to measure changes in active CTNNB1 signaling if that's the route they want to go down.

Authors: We apologize for that. We now provide that information in the Methods section:

“CK1 (Cell Signaling Technology, 1:1000, cat. #2655), Phospho-Akt (Ser473) (Cell Signaling Technology, 1:1000, cat. #4058), Akt (Cell Signaling Technology, 1:1000, cat. #9272), MITF (Cell Signaling Technology, 1:1000, cat. #97800), GSK-3β (Cell Signaling Technology, 1:1000, cat. #9832), CREB and pCREB (Cell Signaling Technology, 1:1000, cat. #9197S and #9198S), Vinculin (Cell Signaling Technology, 1:1000, cat. #4650), GAPDH (Cell Signaling Technology, 1:1000, cat. #5174S), Anti-Mouse (Cell Signaling Technology, 1:10,000, cat. #7076), and Anti-Rabbit (Cell Signaling Technology, 1:10,000, cat. #7074),

GAPDH (Cell Signaling Technology, 1:1000, cat. #5174), GSK-3 β (Cell Signaling Technology, 1:1000, cat. #9832), MITF (Cell Signaling Technology, 1:1000, cat. #97800), Rab7 (Cell Signaling Technology, 1:1000, cat. #2094S), Rab7 (Cell Signaling Technology, 1:1000, cat. #9367S), β -Catenin (Cell Signaling Technology, 1:1000, cat. #9562), β -Actin (Santa Cruz Biotechnology, 1:1000, cat. #Sc-47778), anti-c-myc (Santa Cruz Biotechnology, 1:1000, cat. #sc-40), and anti-RFP (Proteintech, 1:1000, cat. #6g6).”

REVIEWERS' COMMENTS

Reviewer #5 (Remarks to the Author):

The authors have addressed most of my points. I think the manuscript has been modified and nuanced sufficiently.

I do still think they lack direct evidence for their mechanism really being downstream on a WNT signal (as their model shows, they assume it is due to changes in the internalization of the receptor complex that ultimately the levels/activity of GSK3 is also changed) - and while they did do a western blot for phospho and non-phospho CTNNB1 levels, they didn't test the effect of WNT3A stimulation on total, phospho and dephospho CTNNB1 levels (as I suggested). Similarly, we still miss a read out on the transcriptional effects of CTNNB1.

Reviewer #5 (Remarks to the Author):

The authors have addressed most of my points. I think the manuscript has been modified and nuanced sufficiently.

Authors: We thank the Reviewer for acknowledging that he/she thinks that we have “modified and nuanced“ the MS „sufficiently“.

I do still think they lack direct evidence for their mechanism really being downstream on a WNT signal (as their model shows, they assume it is due to changes in the internalization of the receptor complex that ultimately the levels/activity of GSK3 is also changed) - and while they did do a western blot for phospho and non-phospho CTNNB1 levels, they didn't test the effect of WNT3A stimulation on total, phospho and dephospho CTNNB1 levels (as I suggested). Similarly, we still miss a read out on the transcriptional effects of CTNNB1.

Authors: The cartoon was meant to illustrate our hypothesis without stating that Wnt is proven to be involved. But we agree that we should delete Wnt from the cartoon. Accordingly, our new Fig.8 (cartoon below) does now not include the Wnt pathway any longer. We hope that the Reviewer is d'accord with that solution.

Melanoma Risk Factors

Family History

UVR Exposure

Lighter Skin, Hair, & Eye Color

Moles

Radiation Exposure

Immunodeficiency

TPC2 GOF?

High Rab7a Expression?

WT

Rab7 KO TPC2 KO